# CAViAR: Critic-Augmented Video Agentic Reasoning

## Abstract

Video understanding has seen significant progress in recent years, with models' performance on perception from short clips continuing to rise. Yet, multiple recent benchmarks, such as LVBench, Neptune, and ActivityNet-RTL, show performance wanes for tasks requiring complex reasoning on videos as queries grow more complex and videos grow longer. In this work, we ask: can existing perception capabilities be leveraged to successfully perform more complex video reasoning? In particular, we develop a large language model agent given access to video modules as subagents or tools. Rather than following a fixed procedure to solve queries as in previous work such as Visual Programming, ViperGPT, and MoReVQA, the agent uses the results of each call to a module to determine subsequent steps. Inspired by work in the textual reasoning domain, we introduce a critic to distinguish between instances of successful and unsuccessful sequences from the agent. We show that the combination of our agent and critic achieve strong performance on the previously-mentioned datasets.

## 1 Introduction

Advances in video understanding have been propelled by multimodal large-language models (MLLMs) trained end-to-end on visual and text inputs (Hurst et al., 2024; Team et al., 2023; 2024; Bai et al., 2025). While these systems have made major strides in basic perception, they often falter with queries demanding compositional, multi-step reasoning over long videos (Wang et al., 2024b; Nagrani et al., 2024; Huang et al., 2024). Recently, tool-augmented inference has emerged as a powerful class of models towards achieving compositional reasoning in a a variety of domains (Zhou et al., 2023; Wang et al., 2024a). These methods decompose a query into sub-tasks, invoke specialized modules, and scale to large contexts by selectively "zooming-in" on the relevant portion of the input given appropriate tools. These modular inference approaches offer (i) interpretable decision chains, (ii) graceful scaling as context length and task complexity grow, and (iii) a natural substrate for inference-time reasoning, as they generate structured reasoning traces by construction.

These approaches have not scaled to the complexity and size of video reasoning tasks in large part because they lack the ability to adapt their procedures during execution. Current tool-based methods typically generate fixed procedures (Gupta and Kembhavi, 2023; Surís et al., 2023) or use hand-designed stages (Min et al., 2024) that are directly executed. Such fixed plans require meticulous prompt engineering of API declarations and in-context examples, as the plan cannot be changed once execution starts. This rigidity requires models to stitch together module outputs that they never actually observe, and systems must decide which tool to use when (and with what inputs) without having seen any cases of where the tools succeed or fail. Consequently, one poor decision snowballs into unrecoverable errors and propagates hallucinations unchecked (Tong et al., 2024; Stanić et al., 2024). Video exacerbates this problem because the potential points of failure increase. Rather than rolling out inference plans and hoping for success, if systems adapted their plans to intermediate results as well as explored different strategies, they could find plans that work better depending on the situation.

We introduce Critic-Augmented Video Agentic Reasoning (CAViAR), which pairs a program-generating *agent* with a natural-language *critic*. The agent iteratively composes video modules into sequences of executable programs, considering the result of each before moving on to the next step, enabling adaptability over the course of reasoning. It thereby generates

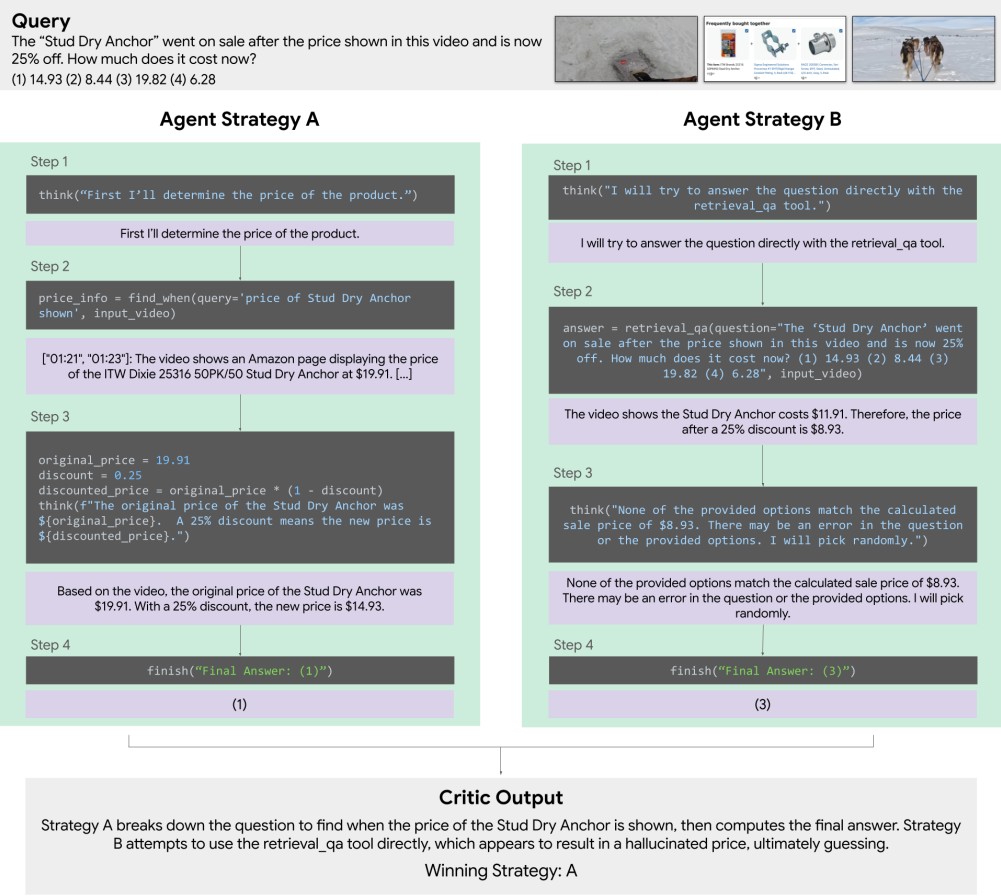

Figure 1: `CAViAR` consists of a reasoning agent that produces sequences of programs to solve video queries with different strategies, followed by a critic that selects the most promising reasoning. Each program invokes visual modules that use the video as input, rather than it being providing as input a single time at the start. We show two strategies here for illustration.

interpretable reasoning traces that lend themselves well to verification, as seen in Agent Strategies A and B in Figure 1. The critic inspects multiple reasoning traces reflecting different solution strategies and selects the most plausible sequence based on examples it has seen, producing natural language feedback as shown by the Critic Output in Figure 1. By comparing multiple strategies, the critic enables success in cases where some approaches lead to success while others fail. Together, the agent and critic allow `CAViAR` to sidestep brittle tool choices and mitigate hallucinations, achieving state-of-the-art results across multiple tasks and datasets.

`CAViAR` has many benefits over previous modular approaches: it is interpretable, as each step corresponds to a short program that can be easily examined; it enables scaling the performance of a single underlying model with no additional training; it avoids issues of customizing module selection to a particular domain, as the critic selects the most promising strategy; it affords compositionality without extreme hand-tuning of module definitions, as the reasoning agent can see module outputs in-context to decide how to use them; and it is general, easily incorporating any additional video modules or Python code that may be useful for a particular domain.

In summary, our contributions are:

1. We present a framework for video understanding using agentic reasoning on video modules.

2. We introduce a critic, which highlights how small amounts of feedback can be used towards substantial performance improvements while avoiding dataset-specific tuning of modules.

3. We show `CAViAR` demonstrates strong performance on multiple tasks such as temporal localization with reasoning and long video question answering on recent, difficult datasets.

## 2 RELATED WORK

**LLMs, Augmented Inference Procedures, and Text Reasoning.** A wide range of work has emerged in recent years using large language models to better solve text reasoning tasks. Cobbe et al. (2021) introduce process supervision, the idea of using information from the model's stated chain-of-thought reasoning to select better answers. Zelikman et al. (2022) filter for chain-of-thought rationales with correct final answers and train the base model further on those. Noting that a correct final answer may not correspond to correct reasoning, Hosseini et al. (2024) build on this by incorporating a verifier trained on pairs of correct and incorrect solutions to predict a scalar value indicating whether a given candidate rationale is correct, using it at inference time to select the best answers. Zhang et al. (2024a) observe that training to produce a scalar value may miss out on the benefits of pretrained LLMs' inherent reasoning capabilities. They point out that the scalar-output verifier approach cannot make use of human-written natural language critiques for why a given candidate answer was wrong rather than just whether it was correct or not, showing that using natural language generation with such critiques to rank reasoning outputs outperforms discriminative scalar-output verifiers. ToRA empowers LLMs with tools for mathematics such as computation libraries callable from Python (e.g., `sympy`), sequentially using code to call tools for math problems. The tool call trajectories are trained to imitate a stronger model, then are filtered for correctness to train the base agent further. These works consider the domain of math word problems, due to its relative ease of verification and the natural breakdown of problems into a structured form of steps and their results.

**Video Understanding and Video Reasoning.** Substantial recent progress in video understanding and video reasoning has come from multimodal language models (Hurst et al., 2024; Team et al., 2023; 2024; Bai et al., 2025). These models are trained with a mixture of video and text inputs to perform next-token prediction of text tokens. SeViLA finetunes the BLIP-2 text-image model to perform localization and QA tasks for video (Yu et al., 2023). Zhang et al. (2024b) demonstrate that synthetic data to copy a stronger model can be a powerful signal for weaker models to understand video. LITA (Huang et al., 2024) finds that despite performance on temporal localization benchmarks, video models struggle to perform localization tasks that requires reasoning. They introduce the ActivityNet-RTL dataset to evaluate this reasoning temporal localization, and show that training with synthetic data from strong teacher models substantially improves performance on this task.

**Agents and Tool Use in Vision.** As the ability of large language models to use tools and perform agentic reasoning has grown, some work has emerged in the visual arena making use of these abilities. Visual Programming (Gupta and Kembhavi, 2023) and ViperGPT (Surís et al., 2023) prompt a language model to produce a program using computer vision tools that solves input queries, philosophically following Johnson et al. (2017), which aimed to perform visual reasoning by learning to generate programs prior to the advent of large language models. Visual Programming relies on many hand-written examples of programs using the given tools. ViperGPT instead defines an API for the provided tools and uses fewer examples, as well as showing some results in the video domain. Further work has found that both the choice of modules per dataset (Khandelwal et al., 2023) and the examples constructed (Stanić et al., 2024) substantially influence the performance of single-program approaches. AVIS (Hu et al., 2023) goes beyond single-program approaches to use tools with tree search for knowledge-intensive image question answering. They define a transition graph of valid tool sequences, which define the tools a language model can choose to use next after each tool. The approach successfully goes beyond fixed procedures, but relies heavily on human knowledge and examples as well as being focused on single images.

Various methods have also emerged recently in this direction specific to video. LLoVi (Zhang et al., 2023) uses a frozen captioner on frames and passes them to an LLM to produce an answer. VideoAgent (Wang et al., 2024c) and VideoTree (Wang et al., 2024d) both use modified inference procedures for video primarily in their selection of frames, both using captions and text similarity. VideoAgent captions every frame and uses an additional frozen image-text similarity model (CLIP (Radford et al., 2021)) to choose frames, iteratively using the captions from those frames with an LLM to produce an answer. They repeat this caption-and-retrieve process, prompting an LLM to ask whether the question has been answered confidently each time until a confidence threshold is reached. VideoTree uses a tree-based representation to build a set of key frames to caption via clustering, then using these captions with an LLM to produce a final answer. MoReVQA (Min et al., 2024) identifies shortcomings in the single-program approach when applied to video, in particular that such planning methods are brittle, failing when inputs do not conform to the expectations set out in

the generated program. They define a three-stage procedure specific to video using visual modules: event parsing, grounding, and reasoning, leading to a final prediction. These fixed stages are used to produce predictions for video queries given few-shot examples of how to use the tools provided.

# 3 METHOD

## 3.1 OVERVIEW

Given a video and a prompt, the reasoning agent iteratively generates and executes programs, using provided video-processing modules and a Python interpreter to ultimately converge to a final answer. We refer to the resulting sequences as reasoning traces or trajectories, as they comprise the full history of steps taken by the reasoning agent to understand the input, reason through the question, and produce a final answer. The reasoning traces produced by the agent are provided to the critic, which provides natural language feedback on their likelihood of success. The feedback from the critic is used to select one of the candidate reasoning traces and its associated final answer. Figure 2 shows the overall procedure. We detail this process in this section.

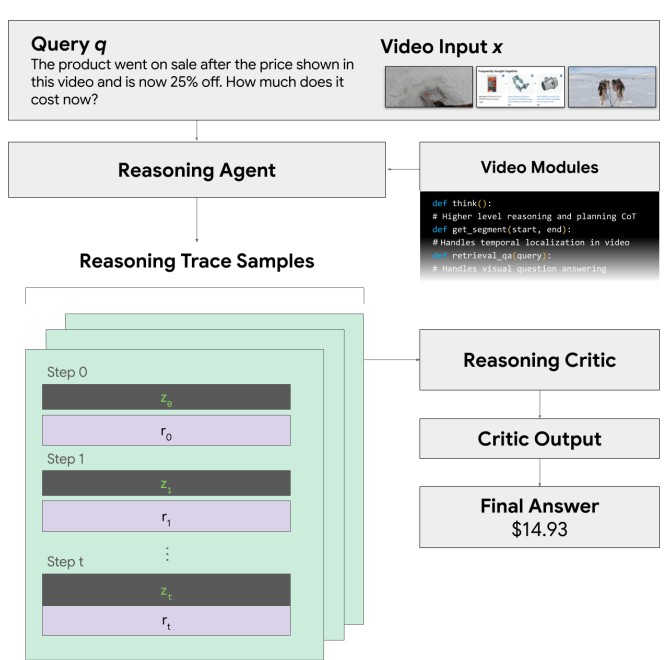

Figure 2: **The `CAViAR` system.** The reasoning agent generates reasoning traces to solve the query using video modules. The critic selects the best, yielding a final answer.

## 3.2 REASONING AGENT

Given a visual (or multimodal) input $x$ and a textual query $q$, `CAViAR` first generates a program $z_1 = \pi(q)$ with the reasoning agent. The reasoning agent $\pi$ is by design fairly simple. We provide it an API for the provided video modules (discussed in the next section) in the form of Python function headers and docstrings.

We note that the exact specifications of outputs need not be as exact as when using a single-program system, as discussed later in this section. The input to the reasoning agent $\pi$, this API (found in full in the Supplementary Material) is comprised of similar definitions for every video module.

The execution engine $\phi$ executes this program on the inputs using the code interpreter and the provided video modules, obtaining the result $r_1 = \phi(x, z_1)$. This already can be a competitive approach in some circumstances, as demonstrated by (Surís et al., 2023; Gupta and Kembhavi, 2023). Yet, the single-program approach has multiple limitations that make its application difficult in practice, as we show in our ablations. It requires careful design of the modules provided, their API descriptions, and the associated examples to ensure the program generator can write programs that use their outputs without being able to see them and decide how to use them next (Stanić et al., 2024). Unlike previous work (Surís et al., 2023; Gupta and Kembhavi, 2023), we explicitly *do not* provide any hand-designed examples of programs or reasoning traces to the agent. While this would likely improve performance, annotating full programs or reasoning traces is challenging and requires expert annotators, which would make the method less usable in practice.

The procedure does not end with the first generated program and result. Rather, the reasoning agent is given both the generated program and its result and asked to produce another program as the next step towards a solution, $z_2 = \pi(q, z_1, r_1)$. This procedure proceeds until the reasoning agent determines it has found a solution, producing a new program $z_i$ based on the full history of program and results and executing it to obtain the next result.

The reasoning agent thus produces a sequence $S = (z_1, r_1, z_2, r_2, \ldots, z_n, r_n)$ culminating in the final answer in $r_n$. (Note that the number of steps, $n$, is not fixed and can be freely decided by the reasoning agent.) We refer to this sequence $S$ as a reasoning trace, or equivalently a trajectory.

## 3.3 VIDEO MODULES AND THEIR API

We aim for a minimal, yet general set of video modules for the experiments we present here. The framework allows for any additional modules to be added on. In all cases, the full range of Python as a programming language is already taken as given – these modules are what we provide on top, thus we do not explicitly note e.g., a 'calculator' tool here. We provide an overview of the modules used and our reasons for including them here, with further implementation details in the Supplementary Material. As a note of terminology, we follow the broader multi-agent literature and refer to the base model being used with different prompts or inference schemes as agents (or subagents) while other operations, such as programmatic ones, are referred to as tools.

**Visual Retrieval + QA (`retrieval_qa`).** One of the most fundamental capabilities needed to understand long videos is the ability to obtain the most visually relevant frames to a given query and use them to get the information corresponding to said query. Retrieval of relevant frames also allows for further intermediate interpretability via visual inspection. In order to use a single model, we perform this retrieval by prompting the model with a sliding window of the frames up to the limit of its context window. At a high level, it uses the visual capabilities of the underlying model to directly ask which individual frames are visually relevant in the given window; the identified frames are then considered to respond to the query. Please see Supplementary Material for further discussion.

**Temporal Grounding (`get_segment`).** Aside from considering the relevant portion of a video based on its visual content, another important ability is to ground temporal information to the video – that is, select the most relevant part of the video based on explicit times. Given a start and end timestamp (and the frame rate for any input video as given), this tool allows the agent to use temporal information in this way by simply trimming out the relevant segment. This can also enable better interpretability in knowing which part of the video the agent chose to consider.

**Temporal Localization (`find_when`).** The natural converse, then, is to identify time information from the video. Given a query corresponding to an event or action, this subagent aims to determine potential ranges of time (identified by a start timestamp and an end timestamp) that may correspond to the given query along with a brief description of what led it to output each range. If the video is longer than the context length available, this subagent also employs a windowed approach, considering the information corresponding to each window and returning any possible ranges found from each window. Its specialized instructions guide it to prioritize recall over precision so as to give the reasoning agent as much information as possible.

**ASR Understanding (`asr_understanding`).** When a video contains speech, it often contains critical context to the visual content of the video. This subagent takes a query and attempts to identify any relevant information from an automatic transcription of the speech in the video. If the transcript is too long for the context, it obtains information from each piece up to its context window, then tries to consolidate the information obtained from the different parts of the transcript.

**Think (`think`).** This basic tool simply allows the reasoning agent to perform a step of explicit verbal reasoning to plan its next tool use before proceeding. It allows the reasoning agent to use a step to reason instead of to use a tool and returns the text reasoning from the agent on how to proceed.

**Completion (`finish`).** The agent indicates its final answer and ends the agentic inference procedure.

## 3.4 REASONING CRITIC

Simply taking the final answer from a single sequence can be a competitive approach, as we later show. Yet, especially when multiple available modules seem like they could lead to a solution, performance can degrade with the single-sequence approach, as we show in our ablations. We note the reasoning agent can make two major categories of errors: faulty reasoning, or tool malfunction.

The reasoning critic, $c$, presents a natural solution to this problem. Rather than sampling a single sequence and accepting the result, we can produce multiple $S_i$, each representing different strategies the reasoning agent can use to obtain an answer. The reasoning critic then critiques the sampled

strategies according to its inherent reasoning capabilities and any examples it has been provided to learn from. Attempting to fact-check from the visual input would raise a chicken-and-egg problem: the critic would in some regards need more reliable visual processing capabilities than the agent itself to make corrections rather than fall for the same hallucinations. In addition, language models have been shown to be poor evaluators of their own predictions without additional information or comparison (Kamoi et al., 2024; Huang et al., 2023); we demonstrate in Section 5 that such self-evaluation results in reduced accuracy. Instead, the critic asks: based on the examples of reasoning traces that have succeeded and failed, how likely is this reasoning trace to succeed? That is, how successful are the reasoning and tool calls given what has worked in the past? Considering these text reasoning traces are also less intensive than entire videos, the feedback from the critic can be used to pick the most promising strategy, as we primarily consider here, or could be used in other ways, such as to provide feedback for further steps from the reasoning agent.

Given reasoning traces $S_i$ corresponding to various sampled strategies, the reasoning critic $c$ aims to provide a critique in the form of natural language feedback for the given strategies; its ultimate goal is to provide a recommendation for which strategy may be the most promising. To obtain trajectories corresponding to markedly different strategies, we provide the reasoning agent different subsets of modules that can lead to a final answer, observing that standard temperature sampling did not produce significant variation in outputs in the video modeling setting. If a module can directly produce an answer of the appropriate type for the query, it is directly applied with guidance to write out its reasoning to provide the critic more information. (For instance: the `retrieval_qa` module produces valid outputs for QA problems while the temporal grounding tool does not.)

Inspired by work in the RLHF space, we note that it is easier both for models and humans to identify a preference between presented options than to assign a well-calibrated numeric score to each independently (Christiano et al., 2017). We therefore present all sampled strategies to the reasoning critic at once, prompting it to critique the given strategies and identify any that could be considered 'winning strategies.' We demonstrate that the critic can achieve strong performance using a small number of in-context examples. Each in-context example is constructed with a question, each sampled strategy, an optional brief critique, and a list of the winning strategies.

## 4 EVALUATION

`CAViAR` can handle a variety of tasks involving multimodal inputs depending on the tools provided. We showcase this with two tasks across three different datasets: two multiple-choice complex long video question answering settings on the LVBench (Wang et al., 2024b) and Neptune (Nagrani et al., 2024) datasets, and the reasoning temporal localization task on the ActivityNet-RTL dataset ((Huang et al., 2024)).(Additional comparison to prior work on the EgoSchema dataset (Mangalam et al., 2023) is presented in the Supplemental Material.)

For our primary results, we consider Gemini Flash 1.5 with 32k token (roughly 120 frames) context as our base model due to a combination of 1) its base ability to process videos 2) its ability to act as an agent given instructions 3) cost, speed, and compute/credit availability considerations. Towards the goal of scaling performance of one model, we implement the critic with the same base model as the agent. To show the generality of our method, we also show some results with GPT-4o-mini in the Supplementary Material. Selecting subsets that result in different strategies leads to 3 strategies for the tasks we consider given the modules described in Section 3.3. We use 4 in-context examples for the critic for each dataset. Full details can be found in the Supplementary Material.

### 4.1 COMPLEX VIDEO QA

First, we explore video question answering. With the rise of powerful video models, difficult benchmarks such as LVBench (Wang et al., 2024b) and Neptune (Nagrani et al., 2024) have emerged, each with unique challenges.

We consider the LVBench dataset (Wang et al., 2024b) (CC-BY-NC-SA-4.0 license) to explore the performance of our method on visually challenging, very long videos. LVBench focuses on visual understanding of particularly long videos, spanning multiple hours. The dataset covers a wide range of domains, including everyday activities, sports, entertainment, and more. Audio and speech information are not allowed, making it a challenging visual-only measure of performance. No

Table 1: **LVBench Results**. We report accuracy on the evaluation set. Direct inference and `CAViAR` results use Gemini 1.5 Flash. Other results reported from dataset leaderboard (Wang et al., 2024b).

| | Accuracy (%) ↑ |
|---|---|
| Kangaroo (Liu et al., 2024) | 38.3 |
| GLM-4V-Plus-0111 (Wang et al., 2023) | 48.7 |
| Qwen2.5-VL (7B) (Bai et al., 2025) | 45.3 |
| Qwen2.5-VL (72B) (Bai et al., 2025) | 47.3 |
| mPLUG-Owl3 (Ye et al., 2024) | 43.5 |
| Direct Inference | 46.0 |
| CAViAR | **62.0** |

Table 2: **Neptune Results**. We report accuracy on the evaluation set. Direct inference and `CAViAR` results use Gemini 1.5 Flash. Other results reported from original dataset (Nagrani et al., 2024).

| | Accuracy (%) ↑ |
|---|---|
| VideoLLaMA2 (Cheng et al., 2024) | 44.7 |
| VideoLLaMA2 with ASR (Cheng et al., 2024) | 49.3 |
| LLaVA-OneVision (Li et al., 2024) | 66.2 |
| InternVL2-8B (Chen et al., 2024) | 57.1 |
| MiniCPM-v (Yao et al., 2024) | 56.6 |
| Direct Inference | 51.4 |
| Direct Inference with ASR | 74.9 |
| CAViAR | **77.2** |

supervised methods are available to compare to as there is no training set. Neptune (Nagrani et al., 2024) (CC-BY/Apache 2.0 license) uses audio information such as speech alongside long videos consisting of many frames. It also covers a wide range of domains, with a focus on varied, complex types of question, such as counting and temporal ordering of events. We evaluate on the full dataset (Neptune-Full).

Results for LVBench can be seen in Table 1. `CAViAR` yields a 13% absolute improvement over the state of the art. Results for Neptune can be seen in Table 2. `CAViAR` yields improvement even over direct inference with ASR provided manually, showing the agentic approach not only successfully integrates the ASR information with the `asr_understanding` module but goes further.

## 4.2 REASONING TEMPORAL LOCALIZATION

The next task we consider is reasoning temporal localization (RTL). Traditional temporal localization aims to identify a time range in a video that corresponds to when a directly visible action or event occurred. Huang et al. (2024) introduce the reasoning time localization task and the ActivityNet-RTL dataset (videos MIT license) with the goal of evaluating queries that require reasoning on top of the localization capabilities to determine an answer. For instance, consider Figure 3. Asking to localize when the woman is communicating with the camera but not interacting with any objects requires not only the direct perception capability to recognize when she is talking to the camera, but also the reasoning capabilities to understand negation and pick the particular instance when she is not

**Query**

When is the woman not seen interacting with any objects but simply talking with the camera?

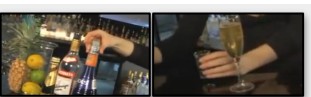

**Critic Selected Reasoning Trace**

Agent Step 1

```
time_ranges = find_when(query="woman talking to camera, not interacting with object")
```

['00:07 - 00:15', '00:52 - 01:06']: From 00:07 to 00:15, a woman is shown talking with the camera without interacting with any objects. From 00:52 to 01:06, she is again shown talking with the camera, holding a cocktail.

Agent Step 2

```
think("The tool find_when returned two time ranges. It indicates that in the first range, she is
talking with the camera without interacting with objects, which fits the question. In the second,
she is holding a cocktail, which is an object. Therefore, only the first is relevant.")
```

The tool find_when returned two time ranges. The description indicates that in [...]

Agent Step 3

```
final_answer = finish("Final answer: ['00:07 - 00:15']")
```

['00:07', '00:15']

Figure 3: **Reasoning temporal localization.** Queries require identifying a time range for events/actions that requires reasoning. Here, the critic chooses a strategy which initially finds multiple ranges but correctly reasons which should be included in a final answer.

interacting with any objects. We include this task to showcase `CAViAR`'s capabilities beyond standard question answering.

We compare to the fully supervised state-of-the-art LITA as well as to the base MLLM. The metric for RTL is mean intersection-over-union (mIOU), the average overlap with the ground truth range over the total union of both across all data points. Table 3 shows the results. `CAViAR` improves on direct usage of the same model by more than 9 points.

Table 3: **ActivityNet-RTL Results**. We report mIOU on the evaluation set. Direct inference results use Gemini 1.5 Flash. Other results reported from (Huang et al., 2024).

|  | mIOU ↑ |
| --- | --- |
| LlaVa-Finetuned (Liu et al., 2023) | 14.6 |
| SlowFast-LlaVa (Huang et al., 2024) | 17.5 |
| LITA (Huang et al., 2024) | 28.6 |
| Direct Inference | 23.0 |
| CAViAR | 32.3 |

## 5  ABLATIONS AND QUALITATIVE RESULTS

We perform ablations to elucidate what makes `CAViAR` work with the LVBench and Neptune datasets.

**Critic vs no critic** One may think on first glance that the reasoning agent armed with video modules could be sufficient on its own. Table 4 shows that not using the critic results in a stark drop in performance, almost to the level of direct inference. Why is this? Looking into the error cases, we find that the `find_when` module appears to be less reliable for very long videos, but given all modules, the agent still tries to use it. This effect is particularly pronounced for LVBench, which has very long videos. See Figure 4 for an example. In this example, the agent tries to use this module to find when the snake discovers the boy, but in its attempt to yield all potentially relevant information, it reports 21 time ranges. For instance, one centers around the boy being found by a scorpion. This module failure ultimately harms the performance of the agent given all modules. The critic sees options sampled with multiple subsets of modules, and is able to pick more accurate strategies than the one the agent picks by default. Prior work analyzing programmatic approaches (Khandelwal et al., 2023) identified their performance is similarly dependent on the set of modules specified per dataset.

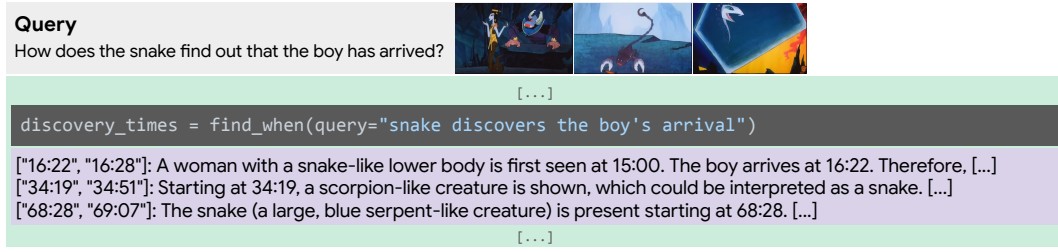

Figure 4: **Module reliability.** The `find_when` module often reports tangentially relevant, distracting information when applied to long videos, often leading the agent with all modules to failure.

**Reasoning agent vs single program** How does generating a single program rather than using the reasoning agent to produce a reasoning trace perform? We present the base model the same API for the modules given to the reasoning agent and instead ask it to generate a single program that solves the query, like in (Surís et al., 2023) or (Gupta and Kembhavi, 2023). As seen in Table 4, using a single program results in nearly random performance in this setting. We find this is due to an overwhelming number of guesses or exceptions from incorrect assumptions of modules when the model tries to write a single program; unlike the agent, a single-program approach is not able to inspect the outputs of a module to see what they actually are like beyond the initial impression from the API. See Figure 5 for a representative example. In this example, the program assumes the answer can be found in the first 15 seconds and that the `retrieval_qa` output will contain an exact answer. As the first 15 seconds do not contain the relevant information, this fails, and the program resorts to a guess. This follows observations from Stanić et al. (2024), who illustrate that single-program approaches are dependent on extensive hand-tuning of API descriptions and expert-annotated examples of successful programs using the provided modules.

**Module configuration.** How much of the critic's performance can be attributed to finding a configuration of modules that works in general rather than per-query? To compare this, we find the optimal

Table 4: **Ablations.**. Items in italics require oracle validation accuracy and are shown only to better understand the method.

| | Accuracy (%) ↑ | |
| --- | --- | --- |
| | LVBench | Neptune |
| Single Program | 27.1 | 28.7 |
| *Single Program (Optimal Fixed Modules)* | 43.3 | 42.0 |
| Agent | 47.1 | 72.5 |
| *Agent (Optimal Fixed Modules)* | 59.8 | 76.5 |
| Agent + Critic | **62.0** | **77.2** |

Table 5: **Self-Evaluation vs Critic.** Using self-evaluation (Wang et al., 2024c) to terminate using confidence scoring *reduces* performance, while the critic substantially increases it.

| | Accuracy (%) ↑ |
| --- | --- |
| Agent | 47.1 |
| Agent + Self-Eval Module (Wang et al., 2024c) | 39.9 |
| Agent + Critic | **62.0** |

fixed subset of modules possible, running each valid combination and considering the evaluation accuracy across the dataset. (Note that rather than oracle per-question accuracy with variable modules, this uses oracle full-dataset accuracy with fixed modules.) For LVBench, it uses both `get_segment` and `retrieval_qa`; for Neptune, `get_segment`, `retrieval_qa`, and `asr_understanding`. Note that this type of module selection typically requires oracle validation, and is therefore not a realistic option in most settings; we present it here to better understand `CAViAR`. Using the optimal subset of modules, the agent performance improves substantially, yet the critic goes even further.

**Query**

What color are the birds at the beginning of the video?

(1) Black (2) Yellow (3) White (4) Blue

**Generated Single Program**

```
segment = get_segment(start='00:00', end='00:15')
# Check the first 15s
answer = retrieval_qa(question, answer_options, segment)
# Assume most likely from first 15s
if "Black" in answer:
  final_answer = "Final Answer: (1)"
elif "Yellow" in answer:
  (...)
else: # Guess if no color mentioned
  final_answer = "Final Answer: (1)"
finish(final_answer)
```

Figure 5: **Modular assumptions hinder single program performance.** In this case, the program assumes the answer appears within the first 15 seconds and that `retrieval_qa` output includes it exactly. Finding no birds in that segment, it lacks viable answers and resorts to a guess.

**Confidence self-evaluation vs critic.** Wang et al. (2024c) introduce a 'confidence' module for deciding when to lock in a final answer. Rather than the agent choosing when to finish and report a final answer on its own as in our work, after each step they prompt an LLM and ask: on a scale of 1-3, how confident are you in this answer? The system repeatedly tries to gather new information until this 'self-evaluation' module reports a confidence of 3, at which point the answer is reported. This could be considered an alternative in some ways to our critic; rather than producing a critique given different strategies based on examples as our critic does, it critiques a single strategy based on the LLM's prior knowledge and ability to self-evaluate. We consider this comparison on LVBench in Table 5. We find that in fact, using self-evaluation leads to a substantial *drop* relative to the agent deciding when to produce an answer on its own. We find this stems from poor calibration of the self-evaluation scores, for instance giving low confidence scores until repeatedly gathering irrelevant information. This follows work showing LLMs are not good judges of their own reasoning in isolation (Kamoi et al., 2024; Huang et al., 2023).

## 6 CONCLUSION

In this work, we present `CAViAR`, a framework for video understanding with agentic reasoning on video modules augmented by reasoning critics. It uses an LLM-driven agent to write sequences of programs, executed using visual modules. The critic selects the most promising strategies, which we show allows for flexible use of modules without hand-customized module selection per dataset. `CAViAR` thus uses additional compute with the same base model to improve performance. Overall, we show `CAViAR` achieves strong performance on multiple reasoning-intensive video tasks.

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

# A APPENDIX

You may include other additional sections here.

