# CAViAR: Critic-Augmented Video Agentic Reasoning Supplemental Material

## 1 Module API

The full module API can be seen in 1. The modules included vary by the strategy being sampled.

```python
@dataclasses.dataclass
class VideoSegment:
  """Class containing a segment of a video, defined by start and end
      times as MM:SS strings."""
  start: str
  end: str

def think(thought: str) -> None:
  """Tool to perform intermediate reasoning that doesn't directly look
      at the video.

  Args:
    thought (str): the thought to print.

  Returns:
    None
  """

def get_segment(start: str, end: str) -> VideoSegment:
  """Clips the segment of the input video between the times indicated
      by 'start' and 'end'.

  Args:
    start (str): the start time for the segment as an MM:SS string.
    end (str): the end time for the segment as an MM:SS string.

  Returns:
    A VideoSegment made from the input video from 'start' to 'end'.
  """

def find_when(query: str, video_segment: VideoSegment | None) -> str:
  """Tool to determine timing of a query such as an event or action.
  Tries to find time ranges in the video that may correspond to the
      query. Works best for simple queries.
  The reasoning may be faulty, so you must carefully consider each
      time range and determine if it is relevant to best answering the
       question.

  Args:
    query (str): the event or action to localize.
    video_segment (VideoSegment): if specified, only looks at the
        given segment of the input video.
  Returns:
    A string of time ranges identified as potentially relevant and the
         justification for each time range.
  """
```

```python
def asr_understanding(question: str, answer_options: list[str] | None)
    :
    """Tool to understand the ASR transcript of the video. Considers the
        ASR transcript of the full video and tries to answer the
        question.
    If the question cannot be answered from the transcript, returns any
        potentially relevant information or timestamps based on the
        transcript.
    Args:
      question (str): the question to answer.
      answer_options (list[str]): Optional. If the set of possible
          answers is known, they should be specified here as a list.
    Returns:
      A string containing the predicted answer to the question.
    """

def retrieval_qa(question: str, answer_options: list[str] | None,
    video_segment: VideoSegment | None) -> str:
    """Tool to answer a question from frames selected independently from
        each minute of the video.
    Looks at each minute of the video and tries to identify relevant
        frames.
    Then tries to answer the question using the retrieved frames.

    Args:
      question (str): the question to answer.
      answer_options (list[str]): Optional. If the set of possible
          answers is known, they should be specified here as a list.
      video_segment (VideoSegment): if specified, only looks at the
          given segment of the input video.

    Returns:
      A string containing the predicted answer to the question.
    """

def finish(final_answer: str) -> str:
    """Print final answer and exit.
    After explaining your reasoning, output the final answer in the
        format "Final Answer: (X)" where X is the correct digit choice.
        Never say "unknown" or "unsure", or "None", instead provide your
        most likely guess.

    Args:
      final_answer (str): reasoning followed by final answer.

    Returns:
      The parsed final answer.
    """
```

Listing 1: Module API.

## 2 MODULE IMPLEMENTATION DETAILS

`VideoSegment`. This is a convenience class for identifying segments of the input video by their start and end time.

`get_segment`. This module selects the appropriate segment from the input video given start and end times, returning a `VideoSegment` corresponding to that range.

`find_when`. This corresponds to coarse temporal localization with a sliding window. Using a window of 100 frames, the model is instructed to identify any time ranges that could be relevant for the given query, ultimately providing all of these ranges along with justifications for each of them as a string.

`asr_understanding`. This module uses the ASR transcript of all the speech in the video with timestamps aligned for the words said at each second. If the query is a question that can be answered from the transcript, it tries to provide an answer. If not, it aims to provide any relevant information along with timestamps from the ASR given.

`retrieval_qa`. This module tries to use the base model to retrieve frames that appear visually relevant to a query, then tries to answer the query from them. It uses a sliding window of 64 frames, returning the indices of frames with high visual relevance from each window. Those frames are then retrieved and provided back to the base model, along with 56 uniform frames from the rest of the video for additional context, to produce a response to the query.

`finish`. This module indicates the agent has reached a final answer and ends the agentic inference procedure, returning the final answer.

```
You are a video understanding agent.  Your task is to answer the
    question provided using the available tools.  These tools are
    already imported and accessible to you. You will receive the
    question and answer choices.  You should use the tools iteratively
    , deciding which tool to use next based on the results of the
    previous tool call. Each turn, you should write a call to one tool
    , enclosed in triple backticks. Continue this process until you
    can confidently answer the question, selecting one of the provided
     answer choices.  Do not import any tools; they are already
    available.  Your final response should be one of the provided
    answer choices.

# Step by Step Instructions

1. **Analyze the question:** Carefully read the provided 'question' to
     understand the specific information required.  Identify key
    entities, actions, and relationships mentioned in the question.

2. **Select an initial tool:** Based on your understanding of the
    question from step 1, choose the most appropriate tool from the
    available set of video interaction functions. Consider which tool
    will provide the most relevant information to answer the question
    efficiently.

3. **Execute the selected tool:** Call the chosen tool, providing any
    necessary parameters derived from the question analysis in step 1.

4. **Analyze the tool's output:** Examine the results returned by the
    tool.  This output might be textual descriptions, numerical data,
    or other relevant information about the video.

5. **Evaluate progress:** Based on the tool's output from step 4,
    assess whether you have enough information to confidently answer
    the question. If yes, proceed to step 6. If not, go back to step 2
     and select a different tool or use the same tool with different
    parameters based on the insights gained from the previous tool
    call.  Consider how the previous tool's output informs your choice
     of the next tool.

6. **Formulate the final answer:** Using the information gathered from
     all tool calls, select one of the provided answer choices that
    best answers the 'question'.  Clearly state your chosen answer
    choice.

Tools:
```

Listing 2: The preamble used for the reasoning agent.

## 3 STRATEGY SAMPLING DETAILS

For each dataset, 3 strategies are selected, each corresponding to a different subset of modules. These correspond to the strategies that can feasibly produce an answer for the given task, as well as the information allowed for the dataset (not including the base 'think' and 'finish' tools). For LVBench (and EgoSchema), the strategies are `get_segment` and `retrieval_qa`; `retrieval_qa` directly; and `get_segment`, `retrieval_qa`, and `find_when` (that is, all modules in use). For Neptune, as audio information is permitted, the same strategies are used with the addition of the `asr_understanding` module for the multi-module strategies. For the RTL task, the strategies are `get_segment` and `find_when`; `find_when` only; and `get_segment`, `retrieval_qa`, and `find_when` (again, all modules in use).

## 4 CRITIC PROMPTS

The critic is given the following preamble.

```
You are an expert in assessing and critiquing reasoning about videos.
    Given a question and possible strategies from video reasoning
    agents for solving it, assess which strategy seems the most likely
    to result in a correct answer based on the provided examples.
If multiple strategies produce the same final answer, you may pick
    multiple; you may not pick multiple answers with conflicting final
    answers.
Observe that the reliability of the tools depends on their input; you
    must consider the examples to determine when a given tool is
    likely to hallucinate or give an accurate output, as evidenced by
    the ultimate result.
Produce a critique per the examples followed by "Winning Strategies:"
    followed by a comma separated list of the letters corresponding to
    those strategies.
```

Listing 3: The preamble used for the critic.

The critic is given 4 in-context examples per prompt. These examples are written by the authors with the aim of being in-distribution for the datasets shown.

```
Input:
You will be given a question about a video and four possible answer
    options. Question: At 12:10, what color suit is the host wearing
    ?Possible answer choices:
(1) Navy
(2) Tan
(3) Black
(4) White
Video length: 40:50

Strategy A (retrieval_qa, get_segment):
'''
input_video = get_segment(start='00:00', end='40:50')
'''

'''
segment = get_segment(start='12:10', end='12:20')
answer = retrieval_qa(question='What color suit is the host wearing?
    ', video_segment=segment)
'''
The host is wearing a navy suit.

'''
final_answer = finish(f"The retrieval_qa tool determined the host
    was wearing a navy suit at 12:10. Final Answer: (1)")
'''
```

```
The retrieval_qa tool determined the host was wearing a navy suit at
    12:10. Final Answer: (1)

Strategy B (direct retrieval_qa):
Here's how to solve this:

1. **Locate the timestamp:** Go to the 12:10 mark in the video.

2. **Observe the host's attire:** At this point, the host (Jimmy
    Kimmel) is wearing a dark suit and a dark-colored tie.

3. **Compare to options:**  The suit appears to be black.

Therefore, the correct answer is (3) Black.
Final Answer: (3)

Strategy C (retrieval_qa, get_segment, find_when):
```
input_video = get_segment(start='00:00', end='40:50')
```

```
segment = get_segment(start='12:10', end='12:20')
```
['12:10', '12:20']

```
answer = retrieval_qa(question='At 12:10, what color suit is the
    host wearing?', answer_options=['Navy', 'Tan', 'Black', 'White'
    ], video_segment=segment)
```
The image at 12:10 shows the host wearing a navy suit.

Final Answer: (1)

```
final_answer = finish(final_answer='The image at 12:10 shows the
    host wearing a navy suit. Final Answer: (1)')
```
The image at 12:10 shows the host wearing a navy suit. Final Answer:
    (1)

Critique:
A: 1. B: 3. C: 1. Strategies A and C succeed while Strategy B fails
    because Strategies A and C successfully use the get_segment tool
     to isolate the relevant part of the video.
Strategy B fails as the retrieval_qa tool alone cannot actually
    locate a given timestamp in a video, resulting in a hallucinated
     answer.

Winning Strategies:
A, C

Input:
You will be given a question about a video and four possible answer
    options. Question: What does the man in charcoal suit do in the
    next scene after the scene of a photo of six people end?Possible
     answer choices:
(1) He shows his scars
(2) He takes off his shirt
(3) He shows his jewelry
(4) He shows his physique
Video length: 39:59
```

```
Strategy A (retrieval_qa, get_segment):
```
input_video = get_segment(start='00:00', end='39:59')
```

```
think("I need to identify the scene where a photo of six people is
    shown. Then I need to find the scene immediately following it
    to see what the man in the charcoal suit does.")
```
I need to identify the scene where a photo of six people is shown.
    Then I need to find the scene immediately following it to see
    what the man in the charcoal suit does.

```
segment_with_photo = get_segment(start='00:00', end='01:00')
answer = retrieval_qa(question="What does the man in charcoal suit
    do in the scene immediately after the scene with a photo of six
    people?", video_segment=segment_with_photo)
```
After the scene with the photo of six people, the man in the
    charcoal suit walks down a hallway and is greeted by a group of
    people who clap and cheer for him.

```
think("The provided text doesn't describe any of the options (1) He
    shows his scars, (2) He takes off his shirt, (3) He shows his
    jewelry, (4) He shows his physique. The man is celebrated. I
    will guess the closest option, which is none of them, but I will
     choose the least violent option.")
```
The provided text doesn't describe any of the options (1) He shows
    his scars, (2) He takes off his shirt, (3) He shows his jewelry,
     (4) He shows his physique. The man is celebrated. I will
    guess the closest option, which is none of them, but I will
    choose the least violent option.

```
final_answer = finish("Final Answer: (1)")
```
Final Answer: (1)

Strategy B (direct retrieval_qa):
Here's how to solve this step-by-step:

1. **Identify the relevant scene:** The question refers to the scene
    after a photo of six people. Looking through the timestamps,
    this appears to be around 20:49.

2. **Observe the next scene:** The next scene shows the man in the
    charcoal suit (Harry Styles) pointing to and showing jewelry on
    his arm.

3. **Match to the answer choices:** This action directly corresponds
    to answer choice (3).

Therefore, the correct answer is **(3) He shows his jewelry**.
Final Answer: (3)

Strategy C (retrieval_qa, get_segment, find_when):
```
input_video = get_segment(start='00:00', end='39:59')
```
```

```
```
think("First, I need to find when the scene with the photo of six
    people ends.  Then I can examine the next scene to see what the
    man in the charcoal suit does.")
```
First, I need to find when the scene with the photo of six people
    ends.  Then I can examine the next scene to see what the man in
    the charcoal suit does.

```
photo_end_time = find_when(query="scene with a photo of six people",
    video_segment=None)
```
["00:05", "00:24"]: The query asks to find the scene with a photo of
     six people.  From the provided frames, a photo of six people is
     visible on the wall in the background starting at 00:05 and
    remaining visible until 00:24.  The photo is partially obscured
    at times by other people and objects, but it is consistently
    present in the background during this timeframe.
["10:39", "10:43"]: The query asks for a scene with a photo of six
    people.  At 10:39 through 10:43 a large group photo of more than
     six people is shown.
["17:47", "17:53"]: The query asks for a scene with a photo of six
    people.  At 17:47-17:53, a photo of approximately six people is
    visible behind a curtain as Harry Styles walks out onto the
    stage.  The photo is partially obscured and the exact number of
    people is difficult to ascertain with certainty, but it appears
    to show at least five.
["18:47", "19:59"]: The query asks for the time range of a scene
    with a photo of six people.  A photo of what appears to be six
    people is visible on the wall behind the main set in frames
    starting at 18:47 and continuing through 19:59.  The photo is
    partially obscured at times by other elements in the scene, but
    it is consistently visible enough to be identified.
["20:03", "20:04"]: The query asks for the time range of a scene
    with a photo of six people.  At 20:03, a wide shot of the studio
     audience is shown, and a photo of six people is visible on a
    screen in the background. This photo remains visible until
    approximately 20:04.
["23:15", "23:25"]: The query asks for the timestamp of a scene with
     a photo of six people.  In the provided video frames, a photo
    of six people is visible on the wall in the background starting
    at 23:15 and remaining visible until 23:25.  The photo is
    partially obscured at times by other elements in the frame, but
    it is consistently present during this time range.
["35:17", "35:20"]: The provided video frames show a wide shot of a
    studio audience at the 35:17 mark.  The audience appears to
    consist of more than six people. While the exact number is
    difficult to determine from the provided frames, it is clearly a
     large group. Therefore, this time range satisfies the query.
["39:03", "39:18"]: The provided video frames show a live
    performance with an audience.  At the end of the performance,
    from 39:03 to 39:18, the camera focuses on the audience, and
    there are enough people visible to potentially include a group
    of five.  The resolution isn't high enough to definitively
    confirm the presence of a specific photo of six people, but the
    scene depicts a large group of people, making it a plausible
    match for the query.

```
think("The most likely candidate for the 'scene with a photo of six
    people' is the longest continuous segment: 18:47-19:59. Let's
    use that as the end point for the previous scene.")
```
```

```
The most likely candidate for the 'scene with a photo of six people'
    is the longest continuous segment: 18:47-19:59. Let's use that
    as the end point for the previous scene.

'''
next_scene_segment = get_segment(start='19:59', end='20:05')
'''
['19:59', '20:05']

'''
answer = retrieval_qa(question="What does the man in the charcoal
    suit do in the scene from 19:59 to 20:05?", answer_options=['He
    shows his scars', 'He takes off his shirt', 'He shows his
    jewelry', 'He shows his physique'], video_segment=
    next_scene_segment)
'''
The provided video frames show a talk show segment featuring three
    men.  From 19:59 to 20:02, the man in the gray suit (Harry
    Styles) sits on a couch and gestures with his hands.  At 20:03,
    the camera angle changes to show the audience, and the three men
     are still on the couch.  There is no indication that the man in
     the gray suit shows his scars, takes off his shirt, shows a
    tattoo, or shows his muscles.  He is simply participating in a
    conversation.

Final Answer: (4)

'''
final_answer = finish("Based on the analysis of the video segment
    from 19:59 to 20:05, the man in the charcoal suit (assuming this
     refers to one of the men in the talk show segment, as there's
    no charcoal suit explicitly mentioned) does not perform any of
    the actions listed in the options. However, since an answer must
     be chosen, and the retrieval_qa tool did not provide a
    definitive answer, I will select the option that is most likely
    to be present in a talk show setting, which is showing his
    physique.  This is a weak inference, but given the constraints,
    it's the best guess. Final Answer: (4)")
'''
Based on the analysis of the video segment from 19:59 to 20:05, the
    man in the charcoal suit (assuming this refers to one of the men
     in the talk show segment, as there's no charcoal suit
    explicitly mentioned) does not perform any of the actions listed
     in the options. However, since an answer must be chosen, and
    the retrieval_qa tool did not provide a definitive answer, I
    will select the option that is most likely to be present in a
    talk show setting, which is showing his physique.  This is a
    weak inference, but given the constraints, it's the best guess.
    Final Answer: (4)

Critique:
A:1. B: 3. C: 4. Strategy B succeeds while Strategies A fails
    because Strategy A incorrectly applies the get_segment tool to
    the first minute of the video and only considers that, resulting
     in it not being able to find a correct answer and ultimately
    guessing. Strategy C applies the find_when tool, then trims a
    segment and does not find the relevant information, then
    guessing as well.

Winning Strategies:
B

Input:
```

```
You will be given a question about a video and four possible answer
    options. Question: What is the main reason the vlogger goes out
    again, where she also buys makeup?Possible answer choices:
(1) To go exercise
(2) To eat chicken
(3) To buy coffee mugs
(4) To go to the park
Video length: 30:25

Strategy A (retrieval_qa, get_segment):
‘‘‘
input_video = get_segment(start='00:00', end='30:25')
‘‘‘

‘‘‘
think("I need to find a segment of the video where the vlogger buys
    makeup. I'll try to identify this segment using retrieval_qa,
    and then I can use that information to answer the question about
     why she went out.")
‘‘‘
I need to find a segment of the video where the vlogger buys makeup.
     I'll try to identify this segment using retrieval_qa, and then
     I can use that information to answer the question about why she
     went out.

‘‘‘
answer = retrieval_qa(question="When does the vlogger buy makeup?",
    answer_options=None, video_segment=input_video)
‘‘‘
The vlogger buys makeup at 16:10-16:11 and 17:20-17:25.

‘‘‘
think("The vlogger buys makeup in two segments: 16:10-16:11 and
    17:20-17:25. I'll use retrieval_qa again, focusing on these
    segments, to determine the reason for her outing.")
‘‘‘
The vlogger buys makeup in two segments: 16:10-16:11 and
    17:20-17:25. I'll use retrieval_qa again, focusing on these
    segments, to determine the reason for her outing.

‘‘‘
segment1 = get_segment(start='16:10', end='16:11')
segment2 = get_segment(start='17:20', end='17:25')
reason = retrieval_qa(question="What is the main reason the vlogger
    goes out again, where she also buys makeup?", answer_options=['
    To go exercise', 'To eat chicken', 'To buy coffee mugs', 'To go
    to the park'], video_segment=segment1)
if reason == "": #check if the first segment didn't provide enough
    information
     reason = retrieval_qa(question="What is the main reason the
        vlogger goes out again, where she also buys makeup?",
        answer_options=['To go exercise', 'To eat chicken', 'To buy
        coffee mugs', 'To go to the park'], video_segment=segment2)
‘‘‘
None

‘‘‘
final_answer = finish("Based on the analysis of the video segments
    where the vlogger buys makeup, I was unable to definitively
    determine the main reason for her outing from the provided
    options. However, given the context of beauty product purchases,
     it's less likely she's going to the gym, eating salmon, or
    going to the park. Therefore, I will eliminate those options.
    Final Answer: (3)")
```

```
‘‘‘
Based on the analysis of the video segments where the vlogger buys
    makeup, I was unable to definitively determine the main reason
    for her outing from the provided options. However, given the
    context of beauty product purchases, it's less likely she's
    going to the gym, eating salmon, or going to the park. Therefore
    , I will eliminate those options.  Final Answer: (3)

Strategy B (direct retrieval_qa):
Here's how to solve this step-by-step:

1. **Identify the relevant section:** The question asks about the
    vlogger's second outing, where she buys makeup.  This occurs
    after the gym visit.

2. **Review the frames:** The frames show the vlogger going to the
    park after the gym.  She then goes shopping and buys makeup at
    Kicks.

3. **Select the answer:** The main reason for her second outing is
    to go to the park.

Therefore, the correct answer is (4) To go to the park.
Final Answer: (4)

Strategy C (retrieval_qa, get_segment, find_when):
‘‘‘
input_video = get_segment(start='00:00', end='30:25')
‘‘‘

‘‘‘
think("I need to find when the vlogger goes out and buys makeup.  I'
    ll use 'find_when' to locate that event, then use 'retrieval_qa'
     to determine the reason.")
‘‘‘
I need to find when the vlogger goes out and buys makeup.  I'll use
    'find_when' to locate that event, then use 'retrieval_qa' to
    determine the reason.

‘‘‘
when_out_buying_makeup = find_when(query="vlogger goes out and buys
    makeup", video_segment=input_video)
‘‘‘
["00:01", "01:59"]: The video shows a vlogger waking up in her room,
     doing some laundry, and then seemingly getting ready to go out.
      While there's no explicit depiction of buying makeup, the
    context strongly suggests this is part of her preparation. The
    time range encompasses her actions from waking up to preparing
    laundry, which could be interpreted as part of her routine
    before going out to buy makeup.
["02:19", "02:25"]: The video shows the vlogger in her bathroom at
    02:15, where she picks up a toothbrush and looks at her tablet.
     From 02:19-02:25, she is shown selecting items from a bathroom
    cabinet. This is consistent with the query.
["05:24", "05:31"]: The vlogger is seen leaving her apartment at
    04:02 and returning at 04:48.  At 05:24-05:31, she is seen
    putting away what appears to be a package of makeup on a shelf
    in her bathroom. This is supported by the context of the query
    and the visual of a package being placed on a shelf.
["08:51", "09:16"]: The video shows a vlogger leaving their home at
    08:30, walking outside, and then entering a grocery store at
    08:51.  The vlogger is seen selecting various grocery items,
    including celery, grapes, blueberries, carrots, broccoli, and
    lettuce, between 08:51 and 09:16.  Therefore, the time range
```

```
        reflects the vlogger's grocery shopping trip, which aligns with
        the query's description of buying products.
["14:00", "14:02"]: The query asks about the vlogger buying makeup.
     The provided video frames show the vlogger in their home,
     interacting with items that could be interpreted as makeup (
     frames 14:00-14:02) and later organizing clothes in their closet
      (frames 14:39-15:25). There is no visual evidence of the
     vlogger going out to buy anything. Therefore, only the initial
     interaction with what appears to be makeup is considered
     relevant.
["16:09", "16:18"], ["17:17", "17:25"]: The video shows the vlogger
     going to a store, Kicks, and buying makeup.  The vlogger is seen
      in the store from 16:09 to 16:18. At 17:17-17:25, the vlogger
     unpacks the Kicks bag, revealing makeup.
["18:12", "18:17"]: The query asks for the time range when the
     vlogger goes out and buys makeup.  The provided video frames
     show the vlogger in their room, at their vanity. At 18:12, there
      are shopping bags from the store Kicks on the vanity. This
     strongly suggests that the vlogger has recently returned from a
     shopping trip where they purchased makeup from Kicks.  There is
     no footage of the actual shopping trip itself within this video
     chunk.
["20:25", "20:51"]: The video shows a person preparing coffee.  From
      20:25 to 20:51, the person is shown unpacking two ceramic brown
      mugs. The contents of the mugs are visible and appear to be
     coffee.  Therefore, this time range is identified as the time
     when the vlogger is shown preparing coffee.
["29:20", "29:41"]: The video shows the vlogger in her bathroom at
     28:00-28:18 applying makeup and then brushing her teeth.  At
     29:20-29:41, she is shown painting her nails.  The presence of
     various makeup in both locations supports the query.

'''

reason = retrieval_qa(question="What is the main reason the vlogger
    goes out again, where she also buys makeup?", answer_options=['
    To go exercise', 'To eat chicken', 'To buy coffee mugs', 'To go
    to the park'], video_segment=get_segment(start='16:09', end='
    17:25'))
'''
The vlogger goes to a store and buys coffee mugs. Then she goes to
    another store and buys makeup. Therefore, the main reason she
    goes out again is to buy coffee mugs.

Final Answer: (3)

'''
final_answer = finish("The provided text from 'retrieval_qa'
    indicates the vlogger buys makeup at a store called Kicks.
    However, it also mentions buying coffee mugs. The question asks
    for the *main* reason for going out again. While makeup is
    mentioned, the provided text strongly suggests the primary
    reason for the second outing is to buy coffee mugs. Therefore,
    the answer is (3). Final Answer: (3)")
'''
The provided text from 'retrieval_qa' indicates the vlogger buys
    makeup at a store called Kicks.  However, it also mentions
    buying coffee mugs. The question asks for the *main* reason for
    going out again. While makeup is mentioned, the provided text
    strongly suggests the primary reason for the second outing is to
     buy coffee mugs. Therefore, the answer is (3). Final Answer:
    (3)

Critique:
A: 3. B: 4. C: 3.
```

```
Strategy B succeeds while Strategies A fails because Strategy A
    looks at particular time ranges of the video and concludes it is
     unable to find a correct answer, thus guessing.
While the approach is generally sound, Strategy A gives up upon not
    finding an answer initially. Strategy C looks at a narrow range
    and follows an implication to guess an answer. Thus Strategy B
    is preferred.

Winning Strategy:
B
```

Listing 4: LVBench critic examples.

```
Input:
You will be given a question about a video and five possible answer
    options. Question:  What are the main points regarding proper form
     and breathing during the barbell squat? Possible answer choices:
(1)  Keep your back straight, core engaged, and chest up throughout
    the movement. Inhale as you descend and exhale forcefully as you
    drive up.
(2)  Maintain a slight arch in your back, push your hips back first,
    and keep your knees aligned with your toes. Breathe in during the
    eccentric phase (lowering) and breathe out during the concentric
    phase (rising).
(3)  Focus on pushing through your heels, keep your head down, and
    hold your breath at the bottom of the squat for stability. Exhale
    as you return to the starting position.
(4)  Prioritize depth over weight, allowing your hips to descend below
     your knees. Inhale deeply at the top, hold your breath throughout
     the squat, and exhale at the top.
(5)  Use a wide stance, keep your toes pointed outwards, and focus on
    squeezing your glutes at the top of the movement. Breathe
    naturally throughout the exercise.
Video length: 01:35

Strategy A (get_segment, retrieval_qa, asr_understanding):
```
input_video = get_segment(start='00:00', end='01:35')
```

```
think("I need to understand the main points about form and breathing
    during barbell squats.  I'll use the ASR understanding tool to
    analyze the video transcript. If that doesn't work, I'll try `
    retrieval_qa` on the video frames.")
```
I need to understand the main points about form and breathing during
    barbell squats.  I'll use the ASR understanding tool to analyze
    the video transcript. If that doesn't work, I'll try `retrieval_qa
    ` on the video frames.

```
asr_understanding(question='What are the main points regarding proper
    form and breathing during the barbell squat?', answer_options=['
    (1)  Keep your back straight, core engaged, and chest up
    throughout the movement. Inhale as you descend and exhale
    forcefully as you drive up.', '(2)  Maintain a slight arch in your
     back, push your hips back first, and keep your knees aligned with
     your toes. Breathe in during the eccentric phase (lowering) and
    breathe out during the concentric phase (rising).', '(3)  Focus on
     pushing through your heels, keep your head down, and hold your
    breath at the bottom of the squat for stability. Exhale as you
    return to the starting position.', '(4)  Prioritize depth over
```

```
        weight, allowing your hips to descend below your knees. Inhale
        deeply at the top, hold your breath throughout the squat, and
        exhale at the top.', '(5)  Use a wide stance, keep your toes
        pointed outwards, and focus on squeezing your glutes at the top of
         the movement. Breathe naturally throughout the exercise.'])
```
The video emphasizes maintaining a slight arch in the back
    (00:23-00:27), pushing the hips back first (00:28-00:31), and
    ensuring knees track with the toes (00:32-00:35). For breathing,
    it advises inhaling during the descent and exhaling during the
    ascent (00:55-01:00).

```
think("The ASR transcript supports answer choice (2). The video
    highlights maintaining a slight arch in your back, initiating the
    movement with the hips, aligning knees with toes, and coordinating
     breathing with the phases of the squat.")
```
The ASR transcript supports answer choice (2). The video highlights
    maintaining a slight arch in your back, initiating the movement
    with the hips, aligning knees with toes, and coordinating
    breathing with the phases of the squat.

```
final_answer = finish("Final Answer: (2)")
```
Final Answer: (2)

Strategy B (direct retrieval_qa):
The video primarily shows a demonstration of the barbell squat and
    highlights key points on form.  It doesn't explicitly detail the
    breathing technique. However, general weightlifting principles
    suggest inhaling during the lowering phase and exhaling during the
     lifting phase. Additionally, proper form includes keeping the
    back straight, core engaged, and chest up.

Final Answer: (1)
Strategy C (get_segment, retrieval_qa find_when, asr_understanding):
```
input_video = get_segment(start='00:00', end='01:35')
```

```
think("I need to find the main points regarding form and breathing in
    the barbell squat.  I'll use `asr_understanding` to see if the
    transcript mentions these details.")
```
I need to find the main points regarding form and breathing in the
    barbell squat.  I'll use `asr_understanding` to see if the
    transcript mentions these details.

```
asr_understanding_result = asr_understanding(question="What are the
    main points regarding proper form and breathing during the barbell
     squat?", answer_options=[
    "Keep your back straight, core engaged, and chest up throughout
        the movement. Inhale as you descend and exhale forcefully as
        you drive up.",
    "Maintain a slight arch in your back, push your hips back first,
        and keep your knees aligned with your toes. Breathe in during
        the eccentric phase (lowering) and breathe out during the
        concentric phase (rising).",
```

```
        "Focus on pushing through your heels, keep your head down, and
            hold your breath at the bottom of the squat for stability.
            Exhale as you return to the starting position.",
        "Prioritize depth over weight, allowing your hips to descend below
             your knees. Inhale deeply at the top, hold your breath
            throughout the squat, and exhale at the top.",
        "Use a wide stance, keep your toes pointed outwards, and focus on
            squeezing your glutes at the top of the movement. Breathe
            naturally throughout the exercise."
])
```
The video focuses on maintaining a slight arch in your back
    (0:23-0:27), pushing your hips back first (0:28-0:31), and keeping
     your knees aligned with your toes (0:32-0:35). For breathing, it
    advises to inhale during the descent (eccentric phase) and exhale
    during the ascent (concentric phase) (0:55-1:00).

```
think("The transcript suggests answer (2), highlighting back arch, hip
    movement, knee alignment, and breathing coordinated with the
    squat phases. Let's use 'retrieval_qa' to cross-check this.")
```
The transcript suggests answer (2), highlighting back arch, hip
    movement, knee alignment, and breathing coordinated with the squat
     phases. Let's use 'retrieval_qa' to cross-check this.

```
retrieval_qa_result = retrieval_qa(question="What are the main points
    regarding proper form and breathing during the barbell squat?",
    answer_options=[
      "Keep your back straight, core engaged, and chest up throughout
          the movement. Inhale as you descend and exhale forcefully as
          you drive up.",
      "Maintain a slight arch in your back, push your hips back first,
          and keep your knees aligned with your toes. Breathe in during
          the eccentric phase (lowering) and breathe out during the
          concentric phase (rising).",
      "Focus on pushing through your heels, keep your head down, and
          hold your breath at the bottom of the squat for stability.
          Exhale as you return to the starting position.",
      "Prioritize depth over weight, allowing your hips to descend below
           your knees. Inhale deeply at the top, hold your breath
          throughout the squat, and exhale at the top.",
      "Use a wide stance, keep your toes pointed outwards, and focus on
          squeezing your glutes at the top of the movement. Breathe
          naturally throughout the exercise."
], video_segment=input_video)
```
The video demonstrates proper form for barbell squats, emphasizing
    keeping the back straight, core engaged, and chest up. While
    breathing is not explicitly addressed, general knowledge suggests
    inhaling on the descent and exhaling forcefully on the ascent.

Final Answer: (1)

```
final_answer = finish("The ASR transcript suggests (2), focusing on
    back arch and hip movement. However, 'retrieval_qa' prioritizes a
    straight back and doesn't fully confirm the breathing technique.
    Considering general weightlifting knowledge and the potential for
    ASR misinterpretation, (1) is the more reliable answer. Final
    Answer: (1)")
```
The ASR transcript suggests (2), focusing on back arch and hip
    movement. However, 'retrieval_qa' prioritizes a straight back and

```
    doesn't fully confirm the breathing technique. Considering general
     weightlifting knowledge and the potential for ASR
    misinterpretation, (1) is the more reliable answer. Final Answer:
    (1)

Critique:
A: 2. B: 1. C: 1. Strategies B and C consider the visual content, but
    deny the information can be found in the video and rely on general
     knowledge.

Winning Strategies:
A

Input:
You will be given a question about a video and five possible answer
    options. Question: To prepare this dish, what are the main steps
    involved?Possible answer choices:
(1) The main steps include chopping vegetables, boiling them in water,
     adding spices and herbs, pouring in coconut milk, simmering the
    mixture, and finally garnishing with cilantro.
(2) The main steps include chopping vegetables, sauteing them in a
    skillet, adding spices and dry fruits, pouring in beef broth,
    baking the mixture in the oven, and finally garnishing with green
    and red bell peppers.
(3) The main steps include chopping vegetables, sauteing them in a wok
    , adding milk and herbs, pouring in vegetable broth, simmering the
     mixture, and finally garnishing with parsley.
(4) The main steps include chopping vegetables, sauteing them in a pan
    , adding lentils, pouring in water, adding spices and cilantro,
    simmering the lentils, and finally garnishing with chopped
    tomatoes.
(5) The main steps include chopping fruits, sauteing them in a pan,
    adding spices and herbs, pouring in chicken broth, simmering the
    mixture, and finally garnishing with cilantro.
Video length: 09:43

Strategy A (get_segment, retrieval_qa, asr_understanding):
'''
input_video = get_segment(start='00:00', end='09:43')
'''

'''
answer = asr_understanding(question='To prepare this dish, what are
    the main steps involved?', answer_options=['The main steps include
     chopping vegetables, boiling them in water, adding spices and
    herbs, pouring in coconut milk, simmering the mixture, and finally
     garnishing with cilantro.', 'The main steps include chopping
    vegetables, sauteing them in a skillet, adding spices and dry
    fruits, pouring in beef broth, baking the mixture in the oven, and
     finally garnishing with green and red bell peppers.', 'The main
    steps include chopping vegetables, sauteing them in a wok, adding
    milk and herbs, pouring in vegetable broth, simmering the mixture,
     and finally garnishing with parsley.', 'The main steps include
    chopping vegetables, sauteing them in a pan, adding lentils,
    pouring in water, adding spices and cilantro, simmering the
    lentils, and finally garnishing with chopped tomatoes.', 'The main
     steps include chopping fruits, sauteing them in a pan, adding
    spices and herbs, pouring in chicken broth, simmering the mixture,
     and finally garnishing with cilantro.'])
'''
The main steps are chopping onions and carrots (00:32-01:15), sauteing
     them in olive oil with garlic and ginger (01:23-02:58), adding
    lentils and water (03:12-04:05), adding turmeric and cumin
    (04:30-05:20), simmering until the lentils are tender
```

```
       (05:35-07:10), and finally garnishing with chopped cilantro and
       tomatoes (07:22-08:01).

'''
think("None of the provided options exactly match the steps in the ASR
      transcript. Option (1) is closest since it mentions chopping
      vegetables, adding spices, and simmering. However, it omits key
      ingredients and steps like sauteing.")
'''
None of the provided options exactly match the steps in the ASR
      transcript. Option (1) is closest since it mentions chopping
      vegetables, adding spices, and simmering. However, it omits key
      ingredients and steps like sauteing.

'''
final_answer = finish("Final Answer: (1)")
'''
Final Answer: (1)

Strategy B (direct retrieval_qa):
The video demonstrates a recipe for lentil soup.  The chef chops
      vegetables like onions, carrots, and garlic. They then saute these
       vegetables in a pan with oil. Lentils and water are added,
      followed by spices like cumin and turmeric. The mixture simmers
      until the lentils are cooked through. Finally, it's garnished with
       chopped tomatoes and cilantro.

The closest answer is (4).

Final Answer: (4)
Strategy C (get_segment, retrieval_qa find_when, asr_understanding):
'''
input_video = get_segment(start='00:00', end='09:43')
'''

'''
answer = asr_understanding(question='To prepare this dish, what are
      the main steps involved?', answer_options=['The main steps include
       chopping vegetables, boiling them in water, adding spices and
      herbs, pouring in coconut milk, simmering the mixture, and finally
       garnishing with cilantro.', 'The main steps include chopping
      vegetables, sauteing them in a skillet, adding spices and dry
      fruits, pouring in beef broth, baking the mixture in the oven, and
       finally garnishing with green and red bell peppers.', 'The main
      steps include chopping vegetables, sauteing them in a wok, adding
      milk and herbs, pouring in vegetable broth, simmering the mixture,
       and finally garnishing with parsley.', 'The main steps include
      chopping vegetables, sauteing them in a pan, adding lentils,
      pouring in water, adding spices and cilantro, simmering the
      lentils, and finally garnishing with chopped tomatoes.', 'The main
       steps include chopping fruits, sauteing them in a pan, adding
      spices and herbs, pouring in chicken broth, simmering the mixture,
       and finally garnishing with cilantro.'])
'''
The main steps include chopping onions and carrots (0:35-1:10),
      sauteing them in olive oil (1:15-2:05), adding spices like cumin
      and turmeric (2:10-2:45), then adding lentils and water
      (2:50-3:30). The mixture simmers for some time (3:45-5:00) and is
      finally garnished with cilantro and chopped tomato (5:10-5:45).

'''
think("None of the options perfectly align with the ASR transcript.
      Option (1) is somewhat similar as it mentions chopping vegetables,
```

```
        spices , and simmering . However , it misses crucial details like
        sauteing and the specific ingredients mentioned .")
```
```
None of the options perfectly align with the ASR transcript . Option
    (1) is somewhat similar as it mentions chopping vegetables , spices
    , and simmering . However , it misses crucial details like sauteing
    and the specific ingredients mentioned .
```
```
final_answer = finish (" Final Answer : (1)")
```
```
Final Answer : (1)

Critique :
A: 1. B: 4. C: 1. A and C consider the ASR , but give up after finding
    it doesn 't directly result in an answer .

Winning Strategies :
B

Input :
You will be given a question about a video and five possible answer
    options . Question : In what order does the video discuss the
    following topics : introduction , stock taillight operation ,
    Morimoto XB LED taillight replacement , taillight features , and
    installation and warranty ? Possible answer choices :
(1) Introduction , stock taillight operation , taillight features ,
    Morimoto XB LED taillight replacement , installation and warranty .
(2) Morimoto XB LED taillight replacement , introduction , taillight
    features , stock taillight operation , installation and warranty .
(3) Introduction , Stock taillight operation , Morimoto XB LED taillight
     replacement , taillight features , Installation and warranty .
(4) Introduction , Morimoto XB LED taillight replacement , installation
    and warranty , taillight features , stock taillight operation .
(5) Stock taillight operation , introduction , Morimoto XB LED taillight
     replacement , taillight features , installation and warranty .
Video length : 01:15

Strategy A (get_segment , retrieval_qa , asr_understanding ):
```
```
input_video = get_segment (start='00:00', end='01:15')
```
```
```
```
answer = asr_understanding (question='In what order does the video
    discuss the following topics : introduction , stock taillight
    operation , Morimoto XB LED taillight replacement , taillight
    features , and installation and warranty?', answer_options=['
    Introduction , stock taillight operation , taillight features ,
    Morimoto XB LED taillight replacement , installation and warranty.'
    , 'Morimoto XB LED taillight replacement , introduction , taillight
    features , stock taillight operation , installation and warranty.',
    'Introduction , Stock taillight operation , Morimoto XB LED
    taillight replacement , taillight features , Installation and
    warranty.', 'Introduction , Morimoto XB LED taillight replacement ,
    installation and warranty , taillight features , stock taillight
    operation.', 'Stock taillight operation , introduction , Morimoto XB
     LED taillight replacement , taillight features , installation and
    warranty.'])
```
```
The video first provides an introduction (00:02-00:10). Next , it
    discusses the stock taillight operation (00:11-00:25). Then , it
    details the Morimoto XB LED taillight replacement (00:26-00:38).
    Following this , the video describes the taillight features
```

```
    (00:39 -00:55). Finally , it covers installation and warranty
    (00:56 -01:08). Therefore , the correct order is (3).

'''
final_answer = finish("Final Answer: (3)")
'''
Final Answer: (3)

Strategy B (direct retrieval_qa):
The video begins with an introduction of Morimoto and then shows the
    stock taillight operation. After that , it shows the Morimoto XB
    LED taillight replacement and then discusses its features. Finally
    , it shows the installation process and discusses its warranty.
    Therefore , the correct order is introduction , stock taillight
    operation , Morimoto XB LED taillight replacement , taillight
    features , and installation and warranty.

Final Answer: (1)
Strategy C (get_segment , retrieval_qa find_when , asr_understanding ):
'''
input_video = get_segment(start='00:00', end='01:15')
'''

'''
answer = asr_understanding(question='In what order does the video
    discuss the following topics: introduction , stock taillight
    operation , Morimoto XB LED taillight replacement , taillight
    features , and installation and warranty?', answer_options=['
    Introduction , stock taillight operation , taillight features ,
    Morimoto XB LED taillight replacement , installation and warranty.'
    , 'Morimoto XB LED taillight replacement , introduction , taillight
    features , stock taillight operation , installation and warranty.',
    'Introduction , Stock taillight operation , Morimoto XB LED
    taillight replacement , taillight features , Installation and
    warranty.', 'Introduction , Morimoto XB LED taillight replacement ,
    installation and warranty , taillight features , stock taillight
    operation.', 'Stock taillight operation , introduction , Morimoto XB
     LED taillight replacement , taillight features , installation and
    warranty.'])
'''
The video first provides an introduction (00:02-00:10). Next, it
    discusses the stock taillight operation (00:11-00:25). Then, it
    details the Morimoto XB LED taillight replacement (00:26-00:38).
    Following this, the video describes the taillight features
    (00:39-00:55). Finally , it covers installation and warranty
    (00:56-01:08). Therefore , the correct order is (3).

'''
final_answer = finish("Final Answer: (3)")
'''
Final Answer: (3)

Critique:
A: 3. B: 1. C: 3. The question asks about what is discussed in the
    video, implying speech, but B is not able to consider the ASR and
    fails.

Winning Strategies:
A, C

Input:
You will be given a question about a video and five possible answer
    options. Question: Why does Jamie decide to stay home instead of
```

```
        going to the library at 11:00 a.m. on a Saturday? Possible answer
        choices:
(1) Jamie gets a call from a friend and decides to meet them for lunch
    instead.
(2) Jamie realizes the library is closed on Saturdays.
(3) Jamie's car breaks down, preventing them from driving to the
    library.
(4) Jamie starts a new book at home and is too engrossed to leave.
(5) Jamie remembers they have a prior engagement, a family gathering,
    preventing them from going to the library.
Video length: 01:59

Strategy A (get_segment, retrieval_qa, asr_understanding):
```
input_video = get_segment(start='00:00', end='01:59')
```

```
answer = asr_understanding(question="Why does Jamie decide to stay
    home instead of going to the library at 11:00 a.m. on a Saturday
    ?", answer_options=['Jamie gets a call from a friend and decides
    to meet them for lunch instead.', 'Jamie realizes the library is
    closed on Saturdays.', 'Jamie\'s car breaks down, preventing them
    from driving to the library.', 'Jamie starts a new book at home
    and is too engrossed to leave.', 'Jamie remembers they have a
    prior engagement, a family gathering, preventing them from going
    to the library.'])
```
Jamie decides to stay home instead of going to the library because
    they remember they have a prior family gathering
    [01:25:00 -01:32:00].

```
final_answer = finish("Based on the ASR transcript, Jamie decides to
    stay home instead of going to the library because they remember
    they have a prior family gathering. Final Answer: (5)")
```
Based on the ASR transcript, Jamie decides to stay home instead of
    going to the library because they remember they have a prior
    family gathering. Final Answer: (5)

Strategy B (direct retrieval_qa):
The video shows Jamie checking the time on their watch, which reads
    11:00 AM. They then pack a bag with books and prepare to leave the
     house, even putting on their shoes. However, they receive a text
    message and after reading it, unpack their bag and take off their
    shoes. They then proceed to sit on the couch and relax. The video
    doesn't explicitly say why the plans changed, but the text message
     is likely the reason.  The video ends with a short scene of a
    group of people, suggesting a possible social gathering.

Final Answer: (1)

Strategy C (get_segment, retrieval_qa find_when, asr_understanding):
```
input_video = get_segment(start='00:00', end='01:59')
```

```
found_times = find_when(query="Jamie's plans are interrupted",
    video_segment=input_video)
```
```

```
["00:35", "00:45"]: The query asks for the time range when Jamie's
    plans are interrupted. Based on the video and transcript, Jamie's
     plans to go to the library are interrupted when they trip over
    their shoelaces after putting on their shoes. This occurs between
     00:35 and 00:42. The video shows them putting on their shoes,
    tripping, and then sitting down.

```
segment_of_interest = get_segment(start='00:35', end='00:45')
```
['00:35', '00:45']

```
answer = retrieval_qa(question="Why does Jamie decide to stay home
    instead of going to the library at 11:00 a.m. on a Saturday?",
    answer_options=['Jamie gets a call from a friend and decides to
    meet them for lunch instead.', 'Jamie realizes the library is
    closed on Saturdays.', 'Jamie\'s car breaks down, preventing them
    from driving to the library.', 'Jamie starts a new book at home
    and is too engrossed to leave.', 'Jamie remembers they have a
    prior engagement, a family gathering, preventing them from going
    to the library.'], video_segment=segment_of_interest)
```
The video shows Jamie putting on their shoes and then tripping over
    their shoelaces. There is no indication they got a call, realized
     the library was closed, had car trouble, or started a new book.
    The video also does not show a family gathering. Therefore, none
    of the options 1-5 are supported by the video.

However, if we must choose the most likely option, option (3) is the
    most plausible. Jamie tripping could be interpreted as an
    indication of car trouble, albeit a very indirect one.

Final Answer: (3)

```
final_answer = finish(final_answer="The video shows Jamie putting on
    their shoes and then tripping over their shoelaces. There is no
    indication they got a call, realized the library was closed, had
    car trouble, or started a new book. The video also does not show a
     family gathering. Therefore, none of the options 1-5 are
    supported by the video. However, if we must choose the most likely
     option, option (3) is the most plausible. Jamie tripping could be
     interpreted as an indication of car trouble, albeit a very
    indirect one. Final Answer: (3)")
```
The video shows Jamie putting on their shoes and then tripping over
    their shoelaces. There is no indication they got a call, realized
     the library was closed, had car trouble, or started a new book.
    The video also does not show a family gathering. Therefore, none
    of the options 1-5 are supported by the video. However, if we must
     choose the most likely option, option (3) is the most plausible.
    Jamie tripping could be interpreted as an indication of car
    trouble, albeit a very indirect one. Final Answer: (3)

Critique:
A: 5. B: 1. C: 3. B makes a guess after not finding an explicit answer
     in the visuals. C selects an overly narrow range to consider, as
    the reason for the plans being interrupted may not be exactly when
     they are interrupted.

Winning Strategies:
A
```

Listing 5: Neptune critic examples.

```
Input:
Question: When is the trampoline first used in the video?
Video length: 04:23

Strategy A (get_segment, find_when):
'''
input_video = get_segment(start='00:00', end='04:23')
'''

'''
time_ranges = find_when(query='the trampoline gets its first use',
    video_segment=input_video)
'''
["00:45", "01:15"]: The video shows a person assembling a trampoline.
    The trampoline is shown fully assembled at 00:45 and the assembly
    process is shown up to 01:15.
["02:30", "02:55"]: The video shows a person jumping on the trampoline
    . This is the first time the trampoline is shown being used, from
    02:30 to 02:55.

'''
think("The first time the trampoline is used is arguably at 00:45,
    when it is assembled.  However, the question asks when it gets its
     *first use*, which implies actual use by a person. Therefore, the
     relevant time range is 02:30-02:55.")
'''
The first time the trampoline is used is arguably at 00:45, when it is
     assembled.  However, the question asks when it gets its *first
    use*, which implies actual use by a person. Therefore, the
    relevant time range is 02:30-02:55.

'''
final_answer = finish("The question asks when the trampoline gets its
    first use, implying actual use by a person. While the trampoline
    is assembled at 00:45-01:15, its first actual use begins at 02:30
    when a person jumps on it. Therefore, the relevant time range is
    02:30-02:55. Final Answer: [02:30, 02:55]")
'''
The question asks when the trampoline gets its first use, implying
    actual use by a person. While the trampoline is assembled at
    00:45-01:15, its first actual use begins at 02:30 when a person
    jumps on it. Therefore, the relevant time range is 02:30-02:55.
    Final Answer: [02:30, 02:55]

Strategy B (get_segment, find_when, retrieval_qa):
'''
input_video = get_segment(start='00:00', end='04:23')
'''

'''
first_use = find_when(query='first use of the trampoline',
    video_segment=input_video)
'''
["00:45", "01:50"]: The video shows a person assembling a trampoline.
    The first appearance of the trampoline occurs at 00:45, when it's
    partially assembled.  The trampoline is shown in several stages of
     assembly until 01:50.
["03:10", "03:40"]: The video shows the finished trampoline being
    showcased in a backyard. The trampoline is fully assembled and
    ready for use at 03:10, and is shown from various angles until
    03:40.
```

```
'''
think("The first time range [00:45, 01:50] seems to describe the
    trampoline being assembled, but the second range [03:10, 03:40]
    shows the trampoline in its final assembled state, ready for use.
     Therefore, the answer is the second range.")
'''
The first time range [00:45, 01:50] seems to describe the trampoline
    being assembled, but the second range [03:10, 03:40] shows the
    trampoline in its final assembled state, ready for use.  Therefore
    , the answer is the second range.

'''
final_answer = finish("The first time range [00:45, 01:50] seems to
    describe the trampoline being assembled, but the second range
    [03:10, 03:40] shows the trampoline in its final assembled state,
    ready for use.  Therefore, the answer is the second range. Final
    Answer: [03:10, 03:40]")
'''
The first time range [00:45, 01:50] seems to describe the trampoline
    being assembled, but the second range [03:10, 03:40] shows the
    trampoline in its final assembled state, ready for use.  Therefore
    , the answer is the second range. Final Answer: [03:10, 03:40]

Strategy C (find_when):
'''
input_video = get_segment(start='00:00', end='03:53')
'''

'''
query = "When is the trampoline first used in the video?"
result = find_when(query, None)
'''
["00:45", "01:15"]: The video shows a person assembling a trampoline.
    The trampoline is shown fully assembled at 00:45 and the assembly
    process is shown up to 01:15.
["02:30", "02:55"]: The video shows a person jumping on the trampoline
    . This is the first time the trampoline is shown being used, from
    02:30 to 02:55.

'''
think("The first result mentions the trampoline being assembled from
    00:45 to 01:15, but this is assembly, not use. The second result
    shows the trampoline in use from 02:30 to 02:55.  I will choose
    the second range as the answer.")
'''
The first result mentions the trampoline being assembled from 00:36 to
     01:59, but this is assembly, not use. The second result shows the
     trampoline in use from 02:30 to 02:55.  I will choose the second
    range as the answer.

'''
final_answer = "The trampoline's first use is shown from 02:30 to
    02:55. Final Answer: [02:30, 02:55]"
finish(final_answer)
'''
The trampoline's first use is shown from 02:30 to 02:55. Final Answer:
    [02:30, 02:55]

Critique:
Strategies A and C actually perceive a person using the trampoline and
     pick the associated range. Strategy B only sees the finished
    trampoline and guesses based on that, but does not directly see
    the part relevant to the query.
```

```
Winning Strategies:
A, C

Question: When is the woman showing a harder variation of the exercise
    ?
Video length: 01:18

Strategy A (get_segment, find_when):
```
input_video = get_segment(start='00:00', end='01:18')
```

```
harder_exercise_times = find_when(query='woman showing a harder
    variation of the exercise', video_segment=input_video)
```
["00:03", "00:27"], ["00:30", "01:15"]: The video shows a woman
    performing a modified push-up against a wall from 0:03 to 0:27.
    From 0:30 to 1:15, she performs standard push-ups on the floor.
    The query asks for a harder version of the exercise. The standard
    push-up on the floor is considered more challenging than the
    modified version against the wall.

```
think("The tool 'find_when' returned two time ranges: ['00:03', '00:27
    '] and ['00:30', '01:15']. The first range shows a modified push-
    up against a wall, and the second shows standard push-ups. The
    standard push-up is described as the harder variation. Therefore,
    the final answer should be the second range.")
```
The tool 'find_when' returned two time ranges: ['00:03', '00:27'] and
    ['00:30', '01:15']. The first range shows a modified push-up
    against a wall, and the second shows standard push-ups. The
    standard push-up is described as the harder variation. Therefore,
    the final answer should be the second range.

```
final_answer = finish("The tool 'find_when' returned two time ranges:
    ['00:03', '00:27'] and ['00:30', '01:15']. The first range shows a
     modified push-up against a wall, and the second shows standard
    push-ups. The standard push-up is described as the harder
    variation. Therefore, the final answer should be the second range.
     Final Answer: [00:30, 01:15]")
```
The tool 'find_when' returned two time ranges: ['00:03', '00:27'] and
    ['00:30', '01:15']. The first range shows a modified push-up
    against a wall, and the second shows standard push-ups. The
    standard push-up is described as the harder variation. Therefore,
    the final answer should be the second range. Final Answer: [00:30,
     01:15]

Strategy B (get_segment, find_when, retrieval_qa):
```
input_video = get_segment(start='00:00', end='01:18')
```

```
times = find_when(query='woman showing a harder variation of the
    exercise', video_segment=input_video)
```
["00:03", "00:27"], ["00:30", "01:15"]: The video shows a woman
    performing a modified push-up against a wall from 0:03 to 0:27.
```

```
    From 0:30 to 1:15, she performs standard push-ups on the floor.
    The query asks for a harder version of the exercise. The standard
    push-up on the floor is considered more challenging than the
    modified version against the wall.

‘‘‘
think("The tool ‘find_when‘ returned two time ranges: 00:03-00:27 and
    00:30-01:15. The description indicates that 00:30-01:15 shows a
    harder variation of the exercise.  I will use this time range as
    the final answer.")
‘‘‘
The tool ‘find_when‘ returned two time ranges: 00:03-00:27 and
    00:30-01:15. The description indicates that 00:30-01:15 shows a
    harder variation of the exercise.  I will use this time range as
    the final answer.

‘‘‘
final_answer = finish("The tool ‘find_when‘ returned two time ranges:
    00:03-00:27 and 00:30-01:15. The description indicates that
    00:30-01:15 shows a harder variation of the exercise.  I will use
    this time range as the final answer. Final Answer: [00:30, 01:15]"
    )
‘‘‘
The tool ‘find_when‘ returned two time ranges: 00:03-00:27 and
    00:30-01:15. The description indicates that 00:30-01:15 shows a
    harder variation of the exercise.  I will use this time range as
    the final answer. Final Answer: [00:30, 01:15]

Strategy C (find_when):
‘‘‘
input_video = get_segment(start='00:00', end='01:18')
‘‘‘

‘‘‘
think("I need to find when the woman is showing a harder variation of
    the exercise. I'll first try to find any mentions of 'harder' or
    similar terms related to the exercise difficulty.")
‘‘‘
I need to find when the woman is showing a harder variation of the
    exercise. I'll first try to find any mentions of 'harder' or
    similar terms related to the exercise difficulty.

‘‘‘
query_result = find_when(query="harder exercise", video_segment=None)
‘‘‘
["00:03", "00:27"]: The query "harder exercise" is subjective. However
    , based on the video content, the frames from 00:03 to 00:27 show
    a person performing a modified exercise, which could be considered
     an "easier exercise." The exercise is shown for a sustained
    period.  Since a harder exercise was requested, it is likely shown
     at a different time in the video.

‘‘‘
final_answer = finish("Based on the tool's analysis, the woman
    performs a harder version of the exercise between 00:03 and 00:27.
     The tool identified this segment, but it may have mistaken it for
     an easier variation. Final Answer: [00:03, 00:27]")
‘‘‘
Based on the tool's analysis, the woman performs a harder version of
    the exercise between 00:03 and 00:27. The tool identified this
    segment, but it may have mistaken it for an easier variation.
    Final Answer: [00:03, 00:27]

Critique:
```

```
Strategies A and B find ranges corresponding to both versions of the
    exercise and compare them to find the harder variation.
Strategy C only identified one range, for a potentially easier version
    , then did not recognize the alternative to compare to.

Winning Strategies:
A, B

Question: When are we given a zoomed in view of the guitar strings?
Video length: 01:53

Strategy A (get_segment, find_when):
```
input_video = get_segment(start='00:00', end='01:53')
```

```
times = find_when(query='zoomed in view of the guitar strings',
    video_segment=input_video)
```
["00:00", "01:52"]: The video shows a person playing a guitar
    throughout its duration.  The query asks for a zoomed in view of
    the guitar strings. Therefore, the entire video shows the event.

```
final_answer = finish("The query asks for when a zoomed in view of the
     guitar strings is given. The response from `find_when` indicates
    that a guitar is played throughout the entire video. Therefore,
    the entire video is relevant. Final Answer: [00:00, 01:53]")
```
The query asks for when a zoomed in view of the guitar strings is
    given. The response from `find_when` indicates that a guitar is
    played throughout the entire video. Therefore, the entire video is
     relevant. Final Answer: [00:00, 01:53]

Strategy B (get_segment, find_when, retrieval_qa):
```
input_video = get_segment(start='00:00', end='01:53')
```

```
times = find_when(query='zoomed in view of the guitar strings',
    video_segment=input_video)
```
["00:00", "01:52"]: The video shows a person playing a guitar
    throughout its duration.  The query asks for a zoomed in view of
    the guitar strings. Therefore, the entire video shows the event.

```
final_answer = finish(final_answer="The video shows a person playing a
     guitar throughout its duration. The query asks for a zoomed in
    view of the guitar strings. Therefore, the entire video shows the
    event. Final Answer: [00:00, 01:53]")
```
The video shows a person playing a guitar throughout its duration. The
     query asks for a zoomed in view of the guitar strings. Therefore,
     the entire video shows the event. Final Answer: [00:00, 01:53]

Strategy C (find_when):
```
input_video = get_segment(start='00:00', end='01:53')
```
```

```
```
think("I need to find when the video shows a close-up of the guitar
    strings. I'll use the 'find_when' tool to search for this.")
```
I need to find when the video shows a close-up of the guitar strings.
    I'll use the 'find_when' tool to search for this.

```
find_when(query="close-up of guitar strings", video_segment=None)
```
["01:10", "01:25"]:  The video shows a person playing guitar. From
    [01:10] to [01:25], the camera zooms in to show a close-up view of
     the guitar strings as the person plays them.

```
think("The 'find_when' tool returned a single time range: 01:10-01:25.
     This seems to directly answer the question. I will use this as
    my final answer.")
```
The 'find_when' tool returned a single time range: 01:10-01:25.  This
    seems to directly answer the question. I will use this as my final
     answer.

```
finish("The find_when tool identified the time range 01:10-01:25 as
    showing a close-up of the guitar strings. Final Answer: [01:10,
    01:25]")
```
The find_when tool identified the time range 01:10-01:25 as showing a
    close-up of the guitar strings. Final Answer: [01:10, 01:25]

Critique:
Strategies A and B choose the whole video, despite the query asking
    for a zoomed in view, implying a relative change in the video.
Strategy C instead identifies a particular range.

Winning Strategies:
C

Input:
Question: When is the person least energetic in the video?
Video length: 00:49

Strategy A (get_segment, find_when):
```
input_video = get_segment(start='00:00', end='00:49')
```

```
activity_times = find_when(query='person least energetic',
    video_segment=input_video)
```
["00:11", "00:18"], ["00:19", "00:24"], ["00:33", "00:41"]: The query
    asks to identify the time ranges where the person is least
    energetic. Based on the provided video frames, the person appears
    to be sleeping during the periods from 00:11 to 00:18, from 00:19
    to 00:24, and from 00:33 to 00:41. These segments represent
    periods of minimal movement and apparent rest.

```
think("The tool returned multiple time ranges: ['00:11', '00:18'], ['
    00:19', '00:24'], ['00:33', '00:41'].  Since the question asks for
```

```
      the time the person is LEAST energetic , and the first two ranges
      are contiguous , I will combine them into a single range .")
'''
The tool returned multiple time ranges : ['00:11', '00:18'], ['00:19',
      '00:24'], ['00:33', '00:41'].  Since the question asks for the
      time the person is LEAST energetic , and the first two ranges are
      contiguous , I will combine them into a single range .

'''
final_answer = "The person is least energetic from 00:11 to 00:24, and
       from 00:33 to 00:41. Final Answer: [00:11, 00:24], [00:33, 00:41]
      "
finish ( final_answer )
'''
The person is least energetic from 00:11 to 00:24, and from 00:33 to
      00:41. Final Answer: [00:11, 00:24], [00:33, 00:41]

Strategy B (get_segment , find_when , retrieval_qa ):
'''
input_video = get_segment (start ='00:00', end ='00:49')
'''

'''
activity_times = find_when (query ='person is least energetic ',
      video_segment = input_video )
'''
["00:11", "00:18"]: The query asks to identify when a person is least
      energetic in the video. The video shows a person who appears to be
       sleeping between 00:11 and 00:18.  This is the period of least
      visible movement and likely the time of lowest energy.

'''
think ("The find_when tool suggests the person is least energetic
      between 00:11 and 00:18.  This seems reasonable based on the
      description. I will use this as my final answer .")
'''
The find_when tool suggests the person is least energetic between
      00:11 and 00:18.  This seems reasonable based on the description.
      I will use this as my final answer .

'''
final_answer = finish ("Final Answer: [00:11, 00:18]")
'''
Final Answer: [00:11, 00:18]

Strategy C (find_when ):
'''
input_video = get_segment (start ='00:00', end ='00:49')
'''

'''
think ("I need to find when the person is least energetic. I'll use
      find_when to search for inactivity .")
'''
I need to find when the person is least energetic. I'll use find_when
      to search for inactivity .

'''
activity_times = find_when (query ="person is least energetic",
      video_segment = None )
'''
["00:11", "00:18"]: The query asks to identify when a person is least
      energetic in the video. The person appears to be sleeping between
```

```
       00:11 and 00:18, indicating low energy. There may be other periods
           of low activity, but this period of apparent sleep is the
       clearest indication of low energy.

'''
think("The find_when tool suggests the person is least energetic
    between 00:11 and 00:18. This seems like the most obvious period
    of inactivity (sleeping). I will use this as my final answer.")
'''
The find_when tool suggests the person is least energetic between
    00:11 and 00:18. This seems like the most obvious period of
    inactivity (sleeping). I will use this as my final answer.

'''
final_answer = finish("The find_when tool suggests the person is least
        energetic between 00:11 and 00:18. This seems like the most
       obvious period of inactivity (sleeping). Therefore, I'm choosing
       this as the final answer. Final Answer: [00:11, 00:18]")
'''
The find_when tool suggests the person is least energetic between
    00:11 and 00:18. This seems like the most obvious period of
    inactivity (sleeping). Therefore, I'm choosing this as the final
    answer. Final Answer: [00:11, 00:18]

Winning Strategies:
A
```

Listing 6: ActivityNet-RTL critic examples.

```
Input:
You will be given a question about a video and five possible answer
    options, where C refers to the person wearing the camera. Question
    : What is the main action being performed by C in the video, and
    provide specific moments that support your observation.Possible
    answer choices:
(1) The main action performed by C is applying makeup. This is evident
     as C picks up a brush, looks in a mirror, and applies something
    to their face.
(2) C's main action in the video is eating a meal.  This is clearly
    evident because C lifts a fork to their mouth multiple times and
    chews.
(3) The main action performed by C in the video is reading a book.
    This is evident in the fact that C holds a book open, turns pages,
     and moves their eyes across the page.
(4) In the video, the main action performed by C is talking on the
    phone. This is evident because C holds a phone to their ear and
    speaks into it several times.
(5) The main action performed by C in the video is typing on a
    computer.  This is clearly seen as C sits at a desk, places their
    hands on a keyboard, and moves their fingers.
Video length: 01:48

Strategy A (retrieval_qa, get_segment):
'''
input_video = get_segment(start='00:00', end='01:48')
'''

'''
answer = retrieval_qa(question='What is the main action being
    performed by C in the video, and provide specific moments that
    support your observation.', answer_options=['The main action
    performed by C is applying makeup. This is evident as C picks up a
     brush, looks in a mirror, and applies something to their face.',
    'C\'s main action in the video is eating a meal.  This is clearly
```

```
        evident because C lifts a fork to their mouth multiple times and
        chews.', 'The main action performed by C in the video is reading a
         book. This is evident in the fact that C holds a book open, turns
         pages, and moves their eyes across the page.', 'In the video, the
         main action performed by C is talking on the phone. This is
        evident because C holds a phone to their ear and speaks into it
        several times.', 'The main action performed by C in the video is
        typing on a computer.  This is clearly seen as C sits at a desk,
        places their hands on a keyboard, and moves their fingers.'],
        video_segment=input_video)
```
C is clearly typing on a computer throughout the video.  They are
    sitting at a desk with their hands on a keyboard and their fingers
     moving. While there may be other actions present, the consistent
    and primary action is typing.

Final Answer: (5)

```
final_answer = finish(final_answer='C is clearly typing on a computer
    throughout the video.  They are sitting at a desk with their hands
     on a keyboard and their fingers moving. While there may be other
    actions present, the consistent and primary action is typing.\n\
    nFinal Answer: (5)')
```
C is clearly typing on a computer throughout the video.  They are
    sitting at a desk with their hands on a keyboard and their fingers
     moving. While there may be other actions present, the consistent
    and primary action is typing.

Final Answer: (5)

Strategy B (direct retrieval_qa):
C is clearly typing on a computer throughout the video. They are
    sitting at a desk with their hands on a keyboard and their fingers
     are moving.  This is the primary and most consistent action shown
    .

Final Answer: (5)

Strategy C (retrieval_qa, get_segment, find_when):
```
input_video = get_segment(start='00:00', end='01:48')
```

```
think("The question asks about the main action C performs. I'll use `
    find_when` to check for specific actions mentioned in the options,
     like 'applying makeup', 'eating', 'reading', 'talking on the
    phone', and 'typing'. This will help identify the main action.")
```
The question asks about the main action C performs. I'll use `
    find_when` to check for specific actions mentioned in the options,
     like 'applying makeup', 'eating', 'reading', 'talking on the
    phone', and 'typing'. This will help identify the main action.

```
makeup_times = find_when(query='C applies makeup', video_segment=
    input_video)
eating_times = find_when(query='C eats a meal', video_segment=
    input_video)
reading_times = find_when(query='C reads a book', video_segment=
    input_video)
```
```

```
phone_times = find_when(query='C talks on the phone', video_segment=
    input_video)
typing_times = find_when(query='C types on a computer', video_segment=
    input_video)
```
["00:12", "00:18"]: At around 00:12, C appears to pick up a small
    object, which is potentially a phone, and brings it to their ear.
    The interaction with the object continues until about 00:18.
    Although it is not definitively clear whether C is talking, their
    actions suggest they may be on a phone call.

```
think("'find_when' only returned a possible phone call. The limited
    information isn't conclusive. Let's use 'retrieval_qa' to check if
     C is focused on work, which could suggest typing.")
```
'find_when' only returned a possible phone call. The limited
    information isn't conclusive. Let's use 'retrieval_qa' to check if
     C is focused on work, which could suggest typing.

```
work_focused = retrieval_qa(question="Is C focused on work in the
    video?", answer_options=["Yes", "No"], video_segment=input_video)
```
The video shows C sitting at a desk and using a computer. Although it'
    s not shown exactly what they are doing, their posture and actions
     suggest they may be working.

Final Answer: (2)

```
think("The response suggests work at a desk and computer use. That
    supports typing.  However, the short phone interaction is also a
    possibility. Since 'eating' seems unlikely in a work context, I'll
     guess between phone and typing, leaning towards phone due to the
    'find_when' result.")
```
The response suggests work at a desk and computer use. That supports
    typing.  However, the short phone interaction is also a
    possibility. Since 'eating' seems unlikely in a work context, I'll
     guess between phone and typing, leaning towards phone due to the
    'find_when' result.

```
final_answer = finish(final_answer="The 'find_when' result hinted at
    phone use. 'retrieval_qa' suggested work at a desk with a computer
    , which supports 'typing.'  Lacking definitive proof, I'll guess '
    talking on the phone' based on the slight 'find_when' evidence.
    Final Answer: (4)")
```
The 'find_when' result hinted at phone use. 'retrieval_qa' suggested
    work at a desk with a computer, which supports 'typing.'  Lacking
    definitive proof, I'll guess 'talking on the phone' based on the
    slight 'find_when' evidence.  Final Answer: (4)

Critique:
A: 5. B: 5. C: 4. Strategy C attempts an overly complex strategy with
    the find_when tool and ends up guessing due to limited information
    .

Winning Strategies:
A, B

Input:
```

```
You will be given a question about a video and five possible answer
    options, where C refers to the person wearing the camera. Question
    : Describe how C prepared the ingredients and assembled the
    sandwich in the video, noting key steps and transitions.Possible
    answer choices:
(1) C prepared the ingredients by ordering them online and having them
     delivered. They then assembled the sandwich by layering the
    ingredients between two slices of bread.  They added condiments
    and cut the sandwich in half.
(2) C prepared the ingredients by taking pre-sliced ingredients from
    the refrigerator. They then assembled the sandwich by layering the
     ingredients between two slices of bread.  They added condiments
    and cut the sandwich in half.
(3) C prepared the ingredients by chopping vegetables and slicing meat
    . They then assembled the sandwich by layering the ingredients
    between two slices of bread.  They added condiments and cut the
    sandwich in half.
(4) C prepared the ingredients by microwaving pre-packaged meals. They
     then assembled the sandwich by layering the ingredients between
    two slices of bread.  They added condiments and cut the sandwich
    in half.
(5) C prepared the ingredients by going to a restaurant and getting
    takeout. They then assembled the sandwich by layering the
    ingredients between two slices of bread.  They added condiments
    and cut the sandwich in half.
Video length: 02:45

Strategy A (retrieval_qa, get_segment):
```
input_video = get_segment(start='00:00', end='02:45')
```

```
answer = retrieval_qa(question='Describe how C prepared the
    ingredients and assembled the sandwich in the video, noting key
    steps and transitions. ', answer_options=['C prepared the
    ingredients by ordering them online and having them delivered.
    They then assembled the sandwich by layering the ingredients
    between two slices of bread.  They added condiments and cut the
    sandwich in half.', 'C prepared the ingredients by taking pre-
    sliced ingredients from the refrigerator. They then assembled the
    sandwich by layering the ingredients between two slices of bread.
     They added condiments and cut the sandwich in half.', 'C prepared
     the ingredients by chopping vegetables and slicing meat. They
    then assembled the sandwich by layering the ingredients between
    two slices of bread.  They added condiments and cut the sandwich
    in half.', 'C prepared the ingredients by microwaving pre-packaged
     meals. They then assembled the sandwich by layering the
    ingredients between two slices of bread.  They added condiments
    and cut the sandwich in half.', 'C prepared the ingredients by
    going to a restaurant and getting takeout. They then assembled the
     sandwich by layering the ingredients between two slices of bread.
      They added condiments and cut the sandwich in half.'],
    video_segment=input_video)
```
The video shows C assembling a sandwich. They take various ingredients
    , like sliced meat and vegetables, from the refrigerator.  They
    layer these ingredients between two slices of bread, add
    condiments, and then cut the sandwich in half.  There is no
    indication of ordering food online, chopping vegetables,
    microwaving meals, or getting takeout.  The ingredients appear to
    be pre-sliced.

Final Answer: (2)
```

```
final_answer = finish("The video clearly shows C using pre-sliced
    ingredients from the refrigerator. Therefore, the most accurate
    description is option (2). Final Answer: (2)")
```
The video clearly shows C using pre-sliced ingredients from the
    refrigerator. Therefore, the most accurate description is option
    (2). Final Answer: (2)

Strategy B (direct retrieval_qa):
The video shows C making a sandwich. They retrieve pre-sliced
    ingredients, like lettuce, tomato, and meat, from the refrigerator
    . They then assemble the sandwich, add condiments, and cut it in
    half. The ingredients were prepared by being pre-sliced and stored
     in the refrigerator.

Final Answer: (3)

Strategy C (retrieval_qa, get_segment, find_when):
```
input_video = get_segment(start='00:00', end='02:45')
```

```
think("To understand how the ingredients were prepared, I'll use '
    find_when' to look for actions like chopping or retrieving from
    the refrigerator, and then use 'retrieval_qa' to select the best
    answer.")
```
To understand how the ingredients were prepared, I'll use 'find_when'
    to look for actions like chopping or retrieving from the
    refrigerator, and then use 'retrieval_qa' to select the best
    answer.

```
preparation_times = find_when("C preparing the sandwich ingredients",
    input_video)
```
["00:05", "00:40"]: C retrieves pre-sliced sandwich ingredients from
    the refrigerator.

```
answer = retrieval_qa(question="Describe how C prepared the
    ingredients and assembled the sandwich in the video, noting key
    steps and transitions.", answer_options=["C prepared the
    ingredients by ordering them online and having them delivered.
    They then assembled the sandwich by layering the ingredients
    between two slices of bread. They added condiments and cut the
    sandwich in half.", "C prepared the ingredients by taking pre-
    sliced ingredients from the refrigerator. They then assembled the
    sandwich by layering the ingredients between two slices of bread.
     They added condiments and cut the sandwich in half.", "C prepared
     the ingredients by chopping vegetables and slicing meat. They
    then assembled the sandwich by layering the ingredients between
    two slices of bread. They added condiments and cut the sandwich
    in half.", "C prepared the ingredients by microwaving pre-packaged
     meals. They then assembled the sandwich by layering the
    ingredients between two slices of bread. They added condiments
    and cut the sandwich in half.", "C prepared the ingredients by
    going to a restaurant and getting takeout. They then assembled the
     sandwich by layering the ingredients between two slices of bread.
      They added condiments and cut the sandwich in half."],
    video_segment=input_video)
```

```
'''
The video shows C making a sandwich.  They retrieve pre-sliced
    ingredients, like lettuce, tomato, and meat, from the refrigerator
    .  They then assemble the sandwich, add condiments, and cut it in
    half. The ingredients were prepared by being pre-sliced and stored
     in the refrigerator.

Final Answer: (3)

'''
final_answer = finish("Since C takes pre-sliced ingredients from the
    refrigerator, the correct answer is (3). Final Answer: (3)")
'''
Since C takes pre-sliced ingredients from the refrigerator, the
    correct answer is (3). Final Answer: (3)

Critique:
A: 2. B: 3. C: 3. Strategy A correctly identifies that the ingredients
     are pre-sliced from the refrigerator but fails to choose the
    matching answer.

Winning Strategies:
B, C

Input:
You will be given a question about a video and five possible answer
    options. Question: Compare the lighting conditions in the room at
    the start of the video to those near the end. How has the lighting
     evolved over the course of the video? Possible answer choices:
(1) The lighting in the earlier segments of the video is primarily
    provided by a single overhead lamp, while the lighting in the
    latter segments is primarily provided by multiple spotlights.
(2) The lighting in the earlier segments of the video is primarily
    natural light from a window, while the lighting in the latter
    segments is primarily provided by an overhead lamp.
(3) The lighting in the earlier segments of the video is primarily
    provided by multiple spotlights, while the lighting in the latter
    segments is primarily natural light from a window.
(4) The lighting in the earlier segments of the video is primarily
    natural light from a window, while the lighting in the latter
    segments is dimmer, perhaps due to cloud cover or sunset.
(5) The lighting in the earlier segments of the video is dimmer,
    perhaps due to cloud cover or sunset, while the lighting in the
    latter segments is primarily natural light from a window.
Video length: 03:01

Strategy A (retrieval_qa, get_segment):
'''
input_video = get_segment(start='00:00', end='03:01')
'''

'''
think("To compare the lighting in the earlier vs. latter segments, I'
    ll first use retrieval_qa on the first minute and the last minute.
     This should give a general idea of the differences. Then I can
    refine my approach if needed.")
'''
To compare the lighting in the earlier vs. latter segments, I'll first
     use retrieval_qa on the first minute and the last minute.  This
    should give a general idea of the differences. Then I can refine
    my approach if needed.

'''
first_minute_segment = get_segment(start='00:00', end='01:00')
```

```
first_minute_answer = retrieval_qa(question="Describe the lighting in
    this segment.", video_segment=first_minute_segment)

last_minute_segment = get_segment(start='02:00', end='03:00')
last_minute_answer = retrieval_qa(question="Describe the lighting in
    this segment.", video_segment=last_minute_segment)

think(f"First minute: {{first_minute_answer}}. Last minute: {{
    last_minute_answer}}. Now I need to compare these and select the
    best matching answer choice.")
'''
First minute: The lighting appears to be dim, likely from a weak
    overhead light source.. Last minute: The room is brightly lit by
    natural light coming in from a large window.. Now I need to
    compare these and select the best matching answer choice.

'''
think("The first minute shows dim lighting, which could be interpreted
     as cloud cover or sunset. The last minute shows bright natural
    light.  Option (4) seems close, mentioning dimmer lighting
    initially and then a change, but it doesn't specifically mention
    the natural light. However, it's the closest option.")
'''
The first minute shows dim lighting, which could be interpreted as
    cloud cover or sunset. The last minute shows bright natural light.
     Option (4) seems close, mentioning dimmer lighting initially and
     then a change, but it doesn't specifically mention the natural
    light. However, it's the closest option.

'''
final_answer = finish("Reasoning: The initial dim lighting suggests
    cloud cover or sunset as per option (4), even though the later
    segment has bright natural light, not just a generic change. Final
     Answer: (4)")
'''
Reasoning: The initial dim lighting suggests cloud cover or sunset as
    per option (4), even though the later segment has bright natural
    light, not just a generic change. Final Answer: (4)

Strategy B (direct retrieval_qa):
The video starts with dim lighting, likely due to cloud cover or the
    time of day. As the video progresses, the lighting becomes
    brighter and it appears more natural light is entering the room,
    likely from a window. This suggests the sun might have come out
    from behind the clouds or the video was filmed as the day
    progressed.

Final Answer: (5)

Strategy C (retrieval_qa, get_segment, find_when):
'''
input_video = get_segment(start='00:00', end='03:01')
'''

'''
think("To compare the lighting, I'll analyze the first and last
    minutes using find_when and then use retrieval_qa for a more
    detailed comparison.")
'''
```

```
To compare the lighting, I'll analyze the first and last minutes using
    find_when and then use retrieval_qa for a more detailed
    comparison.

```
early_segment = get_segment(start='00:00', end='01:00')
```
['00:00', '01:00']

```
early_lighting = find_when(query="Describe the lighting conditions.",
    video_segment=early_segment)
```
["00:00", "00:59"]: The lighting conditions appear dim, likely due to
    a weak overhead light source or cloud cover outside.

```
late_segment = get_segment(start='02:00', end='03:01')
```
['02:00', '03:01']

```
late_lighting = find_when(query="Describe the lighting conditions.",
    video_segment=late_segment)
```
["02:00", "03:00"]: The room is brightly lit by natural light
    streaming in from a large window.

```
comparison = retrieval_qa(question="Compare the lighting conditions in
    the room at the start of the video to those near the end. How has
    the lighting evolved over the course of the video?",
    answer_options=[
    "The lighting in the earlier segments of the video is primarily
        provided by a single overhead lamp, while the lighting in the
        latter segments is primarily provided by multiple spotlights."
        ,
    "The lighting in the earlier segments of the video is primarily
        natural light from a window, while the lighting in the latter
        segments is primarily provided by an overhead lamp.",
    "The lighting in the earlier segments of the video is primarily
        provided by multiple spotlights, while the lighting in the
        latter segments is primarily natural light from a window.",
    "The lighting in the earlier segments of the video is primarily
        natural light from a window, while the lighting in the latter
        segments is dimmer, perhaps due to cloud cover or sunset.",
    "The lighting in the earlier segments of the video is dimmer,
        perhaps due to cloud cover or sunset, while the lighting in
        the latter segments is primarily natural light from a window."
], video_segment=input_video)
```
The video starts with dim lighting, likely due to cloud cover or the
    time of day. As the video progresses, the lighting becomes
    brighter and it appears more natural light is entering the room,
    likely from a window. This suggests the sun might have come out
    from behind the clouds or the video was filmed as the day
    progressed.

Final Answer: (5)

```
final_answer = finish("Based on the analysis, the early segment has
    dim lighting (potentially due to cloud cover), while the late
```

```
      segment is lit by natural light. This matches option (5). Final
      Answer: (5)")
```
```
Based on the analysis, the early segment has dim lighting (potentially
      due to cloud cover), while the late segment is lit by natural
      light. This matches option (5). Final Answer: (5)

Critique: A: 4. B: 5. C: 5. Strategy A initially uses the valid
      strategy of using segments at the beginning and end, but
      ultimately does not logically connect the information to the
      appropriate answer choice.

Winning Strategies: B, C

Input:
You will be given a question about a video and five possible answer
      options, where C refers to the person wearing the camera. Question
      : What is the primary action performed with the backpack at the
      beginning and the main purpose of using a backpack in the context
      of this video? please highlight the significance of this action in
       the overall travel process.Possible answer choices:
(1) The primary action performed with the backpack at the beginning is
       to open it. the main purpose of using a backpack in the context
      of this video is to store belongings. opening the backpack allows
      for easy access to items needed for the trip.
(2) The primary action performed with the backpack at the beginning is
       to close it. the main purpose of using a backpack in the context
      of this video is to carry belongings. closing the backpack ensures
       that items are secure and do not fall out during travel.
(3) The primary action performed with the backpack at the beginning is
       to put it on. the main purpose of using a backpack in the context
       of this video is to carry belongings. putting the backpack on
      allows for hands-free travel and even weight distribution.
(4) The primary action performed with the backpack at the beginning is
       to pack it. the main purpose of using a backpack in the context
      of this video is to carry belongings. packing the backpack ensures
       that all necessary items are brought on the trip.
(5) The primary action performed with the backpack at the beginning is
       to take it off. the main purpose of using a backpack in the
      context of this video is to carry belongings. taking the backpack
      off allows for a rest and reduces strain on the back.
Video length: 02:48

Strategy A (retrieval_qa, get_segment):
```
```
input_video = get_segment(start='00:00', end='02:48')
```

```
answer = retrieval_qa(question='What is the primary action performed
      with the backpack at the beginning and the main purpose of using a
       backpack in the context of this video? please highlight the
      significance of this action in the overall travel process.',
      answer_options=['The primary action performed with the backpack at
       the beginning is to open it. the main purpose of using a backpack
       in the context of this video is to store belongings. opening the
      backpack allows for easy access to items needed for the trip.', '
      The primary action performed with the backpack at the beginning is
       to close it. the main purpose of using a backpack in the context
      of this video is to carry belongings. closing the backpack ensures
       that items are secure and do not fall out during travel.', 'The
      primary action performed with the backpack at the beginning is to
      put it on. the main purpose of using a backpack in the context of
```

```
        this video is to carry belongings. putting the backpack on allows
        for hands-free travel and even weight distribution.', 'The primary
         action performed with the backpack at the beginning is to pack it
        . the main purpose of using a backpack in the context of this
        video is to carry belongings. packing the backpack ensures that
        all necessary items are brought on the trip.', 'The primary action
         performed with the backpack at the beginning is to take it off.
        the main purpose of using a backpack in the context of this video
        is to carry belongings. taking the backpack off allows for a rest
        and reduces strain on the back.'], video_segment=input_video)
```
The video shows a person packing a backpack at the beginning. They
        carefully organize and place items inside. This is essential for
        travel as it ensures all needed items are brought along. The main
        purpose of the backpack in the video is to carry belongings.

Final Answer: (4)

```
think("The response aligns with option 4, showing the backpack is
        initially packed, crucial for travel to ensure having necessary
        items. The main purpose is indeed carrying belongings.")
```
The response aligns with option 4, showing the backpack is initially
        packed, crucial for travel to ensure having necessary items. The
        main purpose is indeed carrying belongings.

```
final_answer = finish("Final Answer: (4)")
```
Final Answer: (4)

Strategy B (direct retrieval_qa):
The video shows a person packing a backpack at the beginning. They
        carefully organize and place items inside. This is essential for
        travel as it ensures all needed items are brought along. The main
        purpose of the backpack in the video is to carry belongings.

Final Answer: (4)

Strategy C (retrieval_qa, get_segment, find_when):
```
input_video = get_segment(start='00:00', end='02:48')
```

```
think("I'll use `find_when` to identify the initial backpack action
        and `retrieval_qa` to determine its purpose in the video, then
        analyze the options.")
```
I'll use `find_when` to identify the initial backpack action and `
        retrieval_qa` to determine its purpose in the video, then analyze
        the options.

```
initial_action_times = find_when("someone interacts with a backpack",
        input_video)
```
["00:05", "00:25"]:  The video shows a person actively packing a
        backpack. They are placing clothes, toiletries, and other items
        inside the backpack.

```
["01:50", "01:55"]: The person is seen wearing the backpack and
    walking through an airport terminal.

```
purpose = retrieval_qa(question="What is the main purpose of using a
    backpack in the context of this video?", answer_options=[
    "The main purpose of using a backpack in the context of this video
        is to store documents.",
    "The main purpose of using a backpack in the context of this video
        is to carry belongings.",
    "The main purpose of using a backpack in the context of this video
        is to organize toiletries.",
    "The main purpose of using a backpack in the context of this video
        is to conceal valuables.",
    "The main purpose of using a backpack in the context of this video
        is to transport food."
], video_segment=input_video)
```
The video shows a person packing and then carrying a backpack filled
    with personal belongings during their travel. Therefore, the main
    purpose of the backpack is to carry belongings.

Final Answer: (2)

```
final_answer = finish("`find_when` shows the backpack is first packed,
     and `retrieval_qa` confirms it's used to carry belongings.
    Knowing packing is essential for bringing necessary items, option
    (3), putting on the backpack for hands-free carrying, aligns with
    the main purpose and initial action. Final Answer: (3)")
```
`find_when` shows the backpack is first packed, and `retrieval_qa`
    confirms it's used to carry belongings.  Knowing packing is
    essential for bringing necessary items, option (3), putting on the
     backpack for hands-free carrying, aligns with the main purpose
    and initial action. Final Answer: (3)

Critique:
A: 4. B: 4. C: 2. Strategy C correctly finds that the backpack is
    first packed and then used to carry belongings, and reasons that
    the main purpose is to carry belongings, but ultimately
    illogically concludes the purpose is to put the backpack on
    despite this.

Winning Strategies:
A, B
""").strip()

  else:
    CRITIC_EXAMPLES = textwrap.dedent(f"""
  Input:
  You will be given a question about a video and four possible answer
      options. Question: At 12:10, what color suit is the host wearing
      ?Possible answer choices:
  (1) Navy
  (2) Tan
  (3) Black
  (4) White
  Video length: 40:50

  Strategy A (retrieval_qa, get_segment):
  ```
  input_video = get_segment(start='00:00', end='40:50')
  ```
```

```
'''
segment = get_segment(start='12:10', end='12:20')
answer = retrieval_qa(question='What color suit is the host wearing?
   ', video_segment=segment)
'''
The host is wearing a navy suit.

'''
final_answer = finish(f"The retrieval_qa tool determined the host
   was wearing a navy suit at 12:10. Final Answer: (1)")
'''
The retrieval_qa tool determined the host was wearing a navy suit at
   12:10. Final Answer: (1)

Strategy B (direct retrieval_qa):
Here's how to solve this:

1. **Locate the timestamp:** Go to the 12:10 mark in the video.

2. **Observe the host's attire:** At this point, the host (Jimmy
   Kimmel) is wearing a dark suit and a dark-colored tie.

3. **Compare to options:**  The suit appears to be black.

Therefore, the correct answer is (3) Black.
Final Answer: (3)

Strategy C (retrieval_qa, get_segment, find_when):
'''
input_video = get_segment(start='00:00', end='40:50')
'''

'''
segment = get_segment(start='12:10', end='12:20')
'''
['12:10', '12:20']

'''
answer = retrieval_qa(question='At 12:10, what color suit is the
   host wearing?', answer_options=['Navy', 'Tan', 'Black', 'White'
   ], video_segment=segment)
'''
The image at 12:10 shows the host wearing a navy suit.

Final Answer: (1)

'''
final_answer = finish(final_answer='The image at 12:10 shows the
   host wearing a navy suit. Final Answer: (1)')
'''
The image at 12:10 shows the host wearing a navy suit. Final Answer:
   (1)

Critique:
A: 1. B: 3. C: 1. Strategies A and C succeed while Strategy B fails
   because Strategies A and C successfully use the get_segment tool
   to isolate the relevant part of the video.
Strategy B fails as the retrieval_qa tool alone cannot actually
   locate a given timestamp in a video, resulting in a hallucinated
   answer.

Winning Strategies:
```

```
A , C

Input :
You will be given a question about a video and four possible answer
    options . Question : What does the man in charcoal suit do in the
    next scene after the scene of a photo of six people end ? Possible
     answer choices :
(1) He shows his scars
(2) He takes off his shirt
(3) He shows his jewelry
(4) He shows his physique
Video length : 39:59

Strategy A ( retrieval_qa , get_segment ):
' ' '
input_video = get_segment ( start ='00:00' , end ='39:59')
' ' '

' ' '
think (" I need to identify the scene where a photo of six people is
    shown . Then I need to find the scene immediately following it
    to see what the man in the charcoal suit does .")
' ' '
I need to identify the scene where a photo of six people is shown .
    Then I need to find the scene immediately following it to see
    what the man in the charcoal suit does .

' ' '
segment_with_photo = get_segment ( start ='00:00' , end ='01:00')
answer = retrieval_qa ( question =" What does the man in charcoal suit
    do in the scene immediately after the scene with a photo of six
    people ?" , video_segment = segment_with_photo )
' ' '
After the scene with the photo of six people , the man in the
    charcoal suit walks down a hallway and is greeted by a group of
    people who clap and cheer for him .

' ' '
think (" The provided text doesn 't describe any of the options (1) He
    shows his scars , (2) He takes off his shirt , (3) He shows his
    jewelry , (4) He shows his physique . The man is celebrated . I
    will guess the closest option , which is none of them , but I will
     choose the least violent option .")
' ' '
The provided text doesn 't describe any of the options (1) He shows
    his scars , (2) He takes off his shirt , (3) He shows his jewelry ,
     (4) He shows his physique . The man is celebrated . I will
    guess the closest option , which is none of them , but I will
    choose the least violent option .

' ' '
final_answer = finish (" Final Answer : (1)")
' ' '
Final Answer : (1)

Strategy B ( direct retrieval_qa ):
Here 's how to solve this step - by - step :

1. ** Identify the relevant scene :** The question refers to the scene
     after a photo of six people . Looking through the timestamps ,
    this appears to be around 20:49.
```

2. **Observe the next scene:** The next scene shows the man in the
   charcoal suit (Harry Styles) pointing to and showing jewelry on
   his arm.

3. **Match to the answer choices:** This action directly corresponds
   to answer choice (3).

Therefore, the correct answer is **(3) He shows his jewelry**.
Final Answer: (3)

Strategy C (retrieval_qa, get_segment, find_when):
```
input_video = get_segment(start='00:00', end='39:59')
```

```
think("First, I need to find when the scene with the photo of six
    people ends.  Then I can examine the next scene to see what the
    man in the charcoal suit does.")
```
First, I need to find when the scene with the photo of six people
    ends.  Then I can examine the next scene to see what the man in
    the charcoal suit does.

```
photo_end_time = find_when(query="scene with a photo of six people",
    video_segment=None)
```
["00:05", "00:24"]: The query asks to find the scene with a photo of
    six people.  From the provided frames, a photo of six people is
    visible on the wall in the background starting at 00:05 and
    remaining visible until 00:24.  The photo is partially obscured
    at times by other people and objects, but it is consistently
    present in the background during this timeframe.
["10:39", "10:43"]: The query asks for a scene with a photo of six
    people.  At 10:39 through 10:43 a large group photo of more than
    six people is shown.
["17:47", "17:53"]: The query asks for a scene with a photo of six
    people.  At 17:47-17:53, a photo of approximately six people is
    visible behind a curtain as Harry Styles walks out onto the
    stage.  The photo is partially obscured and the exact number of
    people is difficult to ascertain with certainty, but it appears
    to show at least five.
["18:47", "19:59"]: The query asks for the time range of a scene
    with a photo of six people.  A photo of what appears to be six
    people is visible on the wall behind the main set in frames
    starting at 18:47 and continuing through 19:59.  The photo is
    partially obscured at times by other elements in the scene, but
    it is consistently visible enough to be identified.
["20:03", "20:04"]: The query asks for the time range of a scene
    with a photo of six people.  At 20:03, a wide shot of the studio
    audience is shown, and a photo of six people is visible on a
    screen in the background. This photo remains visible until
    approximately 20:04.
["23:15", "23:25"]: The query asks for the timestamp of a scene with
    a photo of six people.  In the provided video frames, a photo
    of six people is visible on the wall in the background starting
    at 23:15 and remaining visible until 23:25.  The photo is
    partially obscured at times by other elements in the frame, but
    it is consistently present during this time range.
["35:17", "35:20"]: The provided video frames show a wide shot of a
    studio audience at the 35:17 mark.  The audience appears to
    consist of more than six people. While the exact number is

```
        difficult to determine from the provided frames, it is clearly a
          large group. Therefore, this time range satisfies the query.
  ["39:03", "39:18"]: The provided video frames show a live
      performance with an audience. At the end of the performance,
      from 39:03 to 39:18, the camera focuses on the audience, and
      there are enough people visible to potentially include a group
      of five. The resolution isn't high enough to definitively
      confirm the presence of a specific photo of six people, but the
      scene depicts a large group of people, making it a plausible
      match for the query.

  ```

  think("The most likely candidate for the 'scene with a photo of six
      people' is the longest continuous segment: 18:47-19:59. Let's
      use that as the end point for the previous scene.")
  ```
  The most likely candidate for the 'scene with a photo of six people'
       is the longest continuous segment: 18:47-19:59. Let's use that
      as the end point for the previous scene.

  ```
  next_scene_segment = get_segment(start='19:59', end='20:05')
  ```
  ['19:59', '20:05']

  ```
  answer = retrieval_qa(question="What does the man in the charcoal
      suit do in the scene from 19:59 to 20:05?", answer_options=['He
      shows his scars', 'He takes off his shirt', 'He shows his
      jewelry', 'He shows his physique'], video_segment=
      next_scene_segment)
  ```
  The provided video frames show a talk show segment featuring three
      men. From 19:59 to 20:02, the man in the gray suit (Harry
      Styles) sits on a couch and gestures with his hands. At 20:03,
      the camera angle changes to show the audience, and the three men
       are still on the couch. There is no indication that the man in
       the gray suit shows his scars, takes off his shirt, shows a
      tattoo, or shows his muscles. He is simply participating in a
      conversation.

  Final Answer: (4)

  ```
  final_answer = finish("Based on the analysis of the video segment
      from 19:59 to 20:05, the man in the charcoal suit (assuming this
       refers to one of the men in the talk show segment, as there's
      no charcoal suit explicitly mentioned) does not perform any of
      the actions listed in the options. However, since an answer must
       be chosen, and the retrieval_qa tool did not provide a
      definitive answer, I will select the option that is most likely
      to be present in a talk show setting, which is showing his
      physique. This is a weak inference, but given the constraints,
      it's the best guess. Final Answer: (4)")
  ```
  Based on the analysis of the video segment from 19:59 to 20:05, the
      man in the charcoal suit (assuming this refers to one of the men
       in the talk show segment, as there's no charcoal suit
      explicitly mentioned) does not perform any of the actions listed
       in the options. However, since an answer must be chosen, and
      the retrieval_qa tool did not provide a definitive answer, I
      will select the option that is most likely to be present in a
      talk show setting, which is showing his physique. This is a
      weak inference, but given the constraints, it's the best guess.
      Final Answer: (4)
```

```
Critique:
A:1. B: 3. C: 4. Strategy B succeeds while Strategies A fails
    because Strategy A incorrectly applies the get_segment tool to
    the first minute of the video and only considers that, resulting
     in it not being able to find a correct answer and ultimately
    guessing. Strategy C applies the find_when tool, then trims a
    segment and does not find the relevant information, then
    guessing as well.

Winning Strategies:
B

Input:
You will be given a question about a video and four possible answer
    options. Question: What is the main reason the vlogger goes out
    again, where she also buys makeup?Possible answer choices:
(1) To go exercise
(2) To eat chicken
(3) To buy coffee mugs
(4) To go to the park
Video length: 30:25

Strategy A (retrieval_qa, get_segment):
'''
input_video = get_segment(start='00:00', end='30:25')
'''

'''
think("I need to find a segment of the video where the vlogger buys
    makeup.  I'll try to identify this segment using retrieval_qa,
    and then I can use that information to answer the question about
     why she went out.")
'''
I need to find a segment of the video where the vlogger buys makeup.
     I'll try to identify this segment using retrieval_qa, and then
     I can use that information to answer the question about why she
     went out.

'''
answer = retrieval_qa(question="When does the vlogger buy makeup?",
    answer_options=None, video_segment=input_video)
'''
The vlogger buys makeup at 16:10-16:11 and 17:20-17:25.

'''
think("The vlogger buys makeup in two segments: 16:10-16:11 and
    17:20-17:25.  I'll use retrieval_qa again, focusing on these
    segments, to determine the reason for her outing.")
'''
The vlogger buys makeup in two segments: 16:10-16:11 and
    17:20-17:25.  I'll use retrieval_qa again, focusing on these
    segments, to determine the reason for her outing.

'''
segment1 = get_segment(start='16:10', end='16:11')
segment2 = get_segment(start='17:20', end='17:25')
reason = retrieval_qa(question="What is the main reason the vlogger
    goes out again, where she also buys makeup?", answer_options=['
    To go exercise', 'To eat chicken', 'To buy coffee mugs', 'To go
    to the park'], video_segment=segment1)
if reason == "": #check if the first segment didn't provide enough
    information
```

```
        reason = retrieval_qa(question="What is the main reason the
            vlogger goes out again, where she also buys makeup?",
            answer_options=['To go exercise', 'To eat chicken', 'To buy
            coffee mugs', 'To go to the park'], video_segment=segment2)
```
None

```
final_answer = finish("Based on the analysis of the video segments
    where the vlogger buys makeup, I was unable to definitively
    determine the main reason for her outing from the provided
    options. However, given the context of beauty product purchases,
     it's less likely she's going to the gym, eating salmon, or
    going to the park. Therefore, I will eliminate those options.
    Final Answer: (3)")
```
Based on the analysis of the video segments where the vlogger buys
    makeup, I was unable to definitively determine the main reason
    for her outing from the provided options. However, given the
    context of beauty product purchases, it's less likely she's
    going to the gym, eating salmon, or going to the park. Therefore
    , I will eliminate those options.  Final Answer: (3)

Strategy B (direct retrieval_qa):
Here's how to solve this step-by-step:

1. **Identify the relevant section:** The question asks about the
    vlogger's second outing, where she buys makeup.  This occurs
    after the gym visit.

2. **Review the frames:** The frames show the vlogger going to the
    park after the gym.  She then goes shopping and buys makeup at
    Kicks.

3. **Select the answer:** The main reason for her second outing is
    to go to the park.

Therefore, the correct answer is (4) To go to the park.
Final Answer: (4)

Strategy C (retrieval_qa, get_segment, find_when):
```
input_video = get_segment(start='00:00', end='30:25')
```

```
think("I need to find when the vlogger goes out and buys makeup.  I'
    ll use `find_when` to locate that event, then use `retrieval_qa`
     to determine the reason.")
```
I need to find when the vlogger goes out and buys makeup.  I'll use
    `find_when` to locate that event, then use `retrieval_qa` to
    determine the reason.

```
when_out_buying_makeup = find_when(query="vlogger goes out and buys
    makeup", video_segment=input_video)
```
["00:01", "01:59"]: The video shows a vlogger waking up in her room,
     doing some laundry, and then seemingly getting ready to go out.
      While there's no explicit depiction of buying makeup, the
    context strongly suggests this is part of her preparation. The
    time range encompasses her actions from waking up to preparing
```

```
        laundry, which could be interpreted as part of her routine
        before going out to buy makeup.
["02:19", "02:25"]: The video shows the vlogger in her bathroom at
    02:15, where she picks up a toothbrush and looks at her tablet.
     From 02:19-02:25, she is shown selecting items from a bathroom
    cabinet. This is consistent with the query.
["05:24", "05:31"]: The vlogger is seen leaving her apartment at
    04:02 and returning at 04:48.  At 05:24-05:31, she is seen
    putting away what appears to be a package of makeup on a shelf
    in her bathroom. This is supported by the context of the query
    and the visual of a package being placed on a shelf.
["08:51", "09:16"]: The video shows a vlogger leaving their home at
    08:30, walking outside, and then entering a grocery store at
    08:51.  The vlogger is seen selecting various grocery items,
    including celery, grapes, blueberries, carrots, broccoli, and
    lettuce, between 08:51 and 09:16.  Therefore, the time range
    reflects the vlogger's grocery shopping trip, which aligns with
    the query's description of buying products.
["14:00", "14:02"]: The query asks about the vlogger buying makeup.
     The provided video frames show the vlogger in their home,
    interacting with items that could be interpreted as makeup (
    frames 14:00-14:02) and later organizing clothes in their closet
     (frames 14:39-15:25). There is no visual evidence of the
    vlogger going out to buy anything. Therefore, only the initial
    interaction with what appears to be makeup is considered
    relevant.
["16:09", "16:18"], ["17:17", "17:25"]: The video shows the vlogger
    going to a store, Kicks, and buying makeup.  The vlogger is seen
     in the store from 16:09 to 16:18. At 17:17-17:25, the vlogger
    unpacks the Kicks bag, revealing makeup.
["18:12", "18:17"]: The query asks for the time range when the
    vlogger goes out and buys makeup.  The provided video frames
    show the vlogger in their room, at their vanity. At 18:12, there
     are shopping bags from the store Kicks on the vanity. This
    strongly suggests that the vlogger has recently returned from a
    shopping trip where they purchased makeup from Kicks.  There is
    no footage of the actual shopping trip itself within this video
    chunk.
["20:25", "20:51"]: The video shows a person preparing coffee.  From
     20:25 to 20:51, the person is shown unpacking two ceramic brown
     mugs. The contents of the mugs are visible and appear to be
    coffee.  Therefore, this time range is identified as the time
    when the vlogger is shown preparing coffee.
["29:20", "29:41"]: The video shows the vlogger in her bathroom at
    28:00-28:18 applying makeup and then brushing her teeth.  At
    29:20-29:41, she is shown painting her nails.  The presence of
    various makeup in both locations supports the query.

'''

reason = retrieval_qa(question="What is the main reason the vlogger
    goes out again, where she also buys makeup?", answer_options=['
    To go exercise', 'To eat chicken', 'To buy coffee mugs', 'To go
    to the park'], video_segment=get_segment(start='16:09', end='
    17:25'))
'''
The vlogger goes to a store and buys coffee mugs. Then she goes to
    another store and buys makeup. Therefore, the main reason she
    goes out again is to buy coffee mugs.

Final Answer: (3)

'''
final_answer = finish("The provided text from 'retrieval_qa'
    indicates the vlogger buys makeup at a store called Kicks.
    However, it also mentions buying coffee mugs. The question asks
```

Table 1: **EgoSchema Results**. We report accuracy on the evaluation subset of 500. Direct inference results use Gemini 1.5 Flash. Other results reported from respective method papers.

|  | Accuracy (%) ↑ |
|---|---|
| SeViLA Yu et al. (2023) | 25.7 |
| LLoVi Zhang et al. (2023) | 57.6 |
| VideoAgent Wang et al. (2024a) | 60.2 |
| VideoTree Wang et al. (2024b) | 66.2 |
| Direct Inference | 70.6 |
| CAViAR | **72.2** |

```
    for the *main* reason for going out again. While makeup is
    mentioned, the provided text strongly suggests the primary
    reason for the second outing is to buy coffee mugs. Therefore,
    the answer is (3). Final Answer: (3)")
```
The provided text from 'retrieval_qa' indicates the vlogger buys
    makeup at a store called Kicks. However, it also mentions
    buying coffee mugs. The question asks for the *main* reason for
    going out again. While makeup is mentioned, the provided text
    strongly suggests the primary reason for the second outing is to
     buy coffee mugs. Therefore, the answer is (3). Final Answer:
    (3)

Critique:
A: 3. B: 4. C: 3.
Strategy B succeeds while Strategies A fails because Strategy A
    looks at particular time ranges of the video and concludes it is
     unable to find a correct answer, thus guessing.
While the approach is generally sound, Strategy A gives up upon not
    finding an answer initially. Strategy C looks at a narrow range
    and follows an implication to guess an answer. Thus Strategy B
    is preferred.

Winning Strategy:
B
```

Listing 7: EgoSchema critic examples.

## 5 ADDITIONAL RESULTS

### 5.1 EGOSCHEMA

We also report results on the EgoSchema dataset Mangalam et al. (2023) for comparison with other approaches.

### 5.2 GPT-4O-MINI

To show the capabilities of our method with other base models, we obtained preliminary results with the GPT-4o-mini model. Due to cost, lack of credit availability, and query limits, we perform this proof of concept experiment on 25% of LVBench. We see a substantial gain in performance.

## REFERENCES

Karttikeya Mangalam, Raiymbek Akshulakov, and Jitendra Malik. Egoschema: A diagnostic benchmark for very long-form video language understanding. *Advances in Neural Information Processing Systems*, 36:46212–46244, 2023.

Table 2: **LVBench GPT-4o-mini Results**. We report accuracy on 25% of the evaluation set. Direct inference results use GPT-4o-mini. Other results reported from respective method papers.

|  | Accuracy (%) ↑ |
|---|---|
| Direct Inference | 43.2 |
| CAViAR | **49.0** |

Xiaohan Wang, Yuhui Zhang, Orr Zohar, and Serena Yeung-Levy. Videoagent: Long-form video understanding with large language model as agent. In *European Conference on Computer Vision*, pages 58–76. Springer, 2024a.

Ziyang Wang, Shoubin Yu, Elias Stengel-Eskin, Jaehong Yoon, Feng Cheng, Gedas Bertasius, and Mohit Bansal. Videotree: Adaptive tree-based video representation for llm reasoning on long videos. *arXiv preprint arXiv:2405.19209*, 2024b.

Shoubin Yu, Jaemin Cho, Prateek Yadav, and Mohit Bansal. Self-chained image-language model for video localization and question answering. *Advances in Neural Information Processing Systems*, 36:76749–76771, 2023.

Ce Zhang, Taixi Lu, Md Mohaiminul Islam, Ziyang Wang, Shoubin Yu, Mohit Bansal, and Gedas Bertasius. A simple llm framework for long-range video question-answering. *arXiv preprint arXiv:2312.17235*, 2023.