# OpenReview forum: "CAViAR: Critic-Augmented Video Agentic Reasoning"
_ICLR.cc/2026/Conference — Submitted to ICLR 2026_

### Official Review · Reviewer_wZWB · 2025-10-24

**Soundness:** 2
**Presentation:** 2
**Contribution:** 2
**Rating:** 4
**Confidence:** 4

**Summary:**

This paper proposes CAViAR (Critic-Augmented Video Agentic Reasoning), a training-free framework for video understanding. It uses an LLM as a reasoning agent that iteratively builds/executes short programs, leveraging many video tools (e.g., temporal grounding, retrieval+QA, ASR understanding), and then uses the LLM as a critic to select among multiple reasoning traces/strategies. On video QA benchmarks, including LVBench, Neptune, and ActivityNet-RTL, CAViAR reports improvements over direct inference and various baselines; ablations show that most of the improvements come from the critic and from agent multi-step reasoning.

**Strengths:**

- Sound framework. A simple yet working training-free agent framework that uses an LLM agent to generate/execute code with many tools for video understanding.

- Good results. The CAViAR system achieves strong results across multiple video QA benchmarks, including LVBench, Neptune, and ActivityNet-RTL.

- Useful ablations. The paper clearly ablates the effects of the critic and multi-step reasoning on the final video QA performance.

**Weaknesses:**

- Limited novelty. Training-free, agentic frameworks for video understanding have already been explored (e.g., VideoAgent, VideoTree). Programmatic tool use has likewise been studied (e.g., ViperGPT). This paper primarily combines an agent framework with code execution. Given the limited novelty, the work would benefit from deeper analysis, e.g., when code execution helps, when it fails, and guidelines for effective system design.

- Missing baselines. Several relevant training-free agentic systems are absent from the comparisons, notably VideoAgent and VideoTree. Including these (ideally under a shared backbone and context budget) would strengthen the empirical case.

- Additional benchmarks. The evaluation focuses largely on older video-QA datasets. It would be valuable to include more recent and challenging benchmarks such as VideoMME, MovieChat, and NextQA.

**Questions:**

See weaknesses.

Can you also report average #tool calls, tokens, latency, and $/query for LVBench/Neptune/RTL (distribution helps)?

---

> ### Author Response · Authors · 2025-11-17
>
> Thank you for your thorough review and constructive feedback. We appreciate your recognition of our sound framework, strong empirical results, and useful ablations. We address your concerns below.
>
> ## 1. Novelty and baselines:
> Please see the note to all reviewers, where we contrast with other superficially-similar frameworks. We also note we do compare to the key agentic contribution of VideoAgent (holding frame selection constant) with the self-confidence module results in Table 5.
>
> ## 2. Additional benchmarks:
> The claim that “the evaluation focuses largely on older video-QA datasets” appears to be a misunderstanding. All three of our main evaluation settings – ActivityNet-RTL, LVBench, and Neptune – are from 2024 and 2025, and are more recent (as well as more challenging) than the alternative datasets named, MovieChat and NextQA.
>
> ## 3. Inference-time compute metrics:
> Please see the note to all reviewers, where we provide additional numbers and discussion of the number of API calls and tool calls.

---

### Official Review · Reviewer_kQk1 · 2025-10-26

**Soundness:** 2
**Presentation:** 2
**Contribution:** 2
**Rating:** 4
**Confidence:** 4

**Summary:**

CAViAR is a training-free agentic system for long-video reasoning that introduces a two-stage approach: an LLM planner/executor generates multiple programmatic strategies using vision and audio tools (frame retrieval, temporal localization, ASR analysis, etc.), and a separate critic LLM compares the resulting execution traces to select the best answer.

A distinct **trace-level selection** step: a separate LLM **critic** compares multiple *executed* tool programs (with timestamps, ASR spans, intermediate rationales) and **chooses** the best evidence chain—going beyond single-run self-confidence or answer-only voting. The contribution is a **preference-over-programs** formulation that operationalizes cross-trajectory comparison as the decision rule, turning tool-grounded reasoning traces themselves—not just final answers—into the object of evaluation and selection.



Key contributions:

* Multi-strategy execution + cross-trace selection: Unlike prior video agents that rely on single-trace self-evaluation, CAViAR generates diverse reasoning paths and uses a critic to compare them—crucial for handling noisy tool outputs in long videos
* Strong empirical gains: Achieves 62.0% on LVBench (+14-35pts over baselines), 77.2% on Neptune, and 32.3 mIOU on ActivityNet-RTL, outperforming both direct inference and supervised methods

**Strengths:**

* Separates planning/execution from a selection stage where a critic LLM compares multiple evidence-bearing traces and picks the winner—more robust than single-trace self-confidence or fixed pipelines (clear methodological originality).
* Consistent, large gains across diverse long-video settings: strong, training-free improvements on vision-only QA, audio-augmented QA (via ASR tool), and temporal localization, with clean ablations isolating the critic’s contribution (high empirical quality).

**Weaknesses:**

* **Novelty**: limited. Meny works explore utilizing LLMs with Video-LMMs in agentic frameworks. The difference is the utilization of two distinct LLMs for generating programs and as a critic.
- **Insufficient benchmark coverage**: Skips Video-MME, LongVideoBench, and MLVU (temporal ordering/counting—Neptune's known failure modes). Doesn't report Neptune's own GEM metric for evidence grounding.
- **No open-model validation**: Zero replication on open LMMs & LLMs despite claiming "training-free portability." Open model results are important to see if OS models can be utilized in this framework.

**Questions:**

* Your paper's core claim rests on the superiority of a separate critic over other selection methods. While the ablation in Table 5 convincingly shows the critic outperforms a confidence-based self-evaluation module, it omits a comparison to simpler ensemble methods
* The evaluation is limited to LVBench, Neptune, and ActivityNet-RTL, omitting other critical long-video benchmarks like Video-MME and MLVU
* The paper claims the framework is "general" and enables performance scaling "with no additional training," yet all primary results depend on a single proprietary model, Gemini 1.5 Flash. Please include more base models to make sure method generalizes

---

> ### Author Response · Authors · 2025-11-17
>
> Thank you for your detailed review and recognition of our contributions, including how “multi-strategy execution [...] is crucial for handling noisy tool outputs in long videos” and “separating planning and execution with a critic LLM [...] is more robust” showing “clear methodological originality” and “high empirical quality.” We address your concerns below.
>
> ## 1. Novelty:
> Please see the note to all reviewers, where we contrast with other superficially-similar frameworks.
>
> ## 2. Benchmark coverage:
> We disagree that lacking specific, particular benchmarks makes the benchmark coverage ‘insufficient’. We consider three recent, difficult benchmarks, which is standard for other similar works, that cover a range of capabilities  – ActivityNet-RTL for localization, LVBench for long video understanding, Neptune for multimodal – with additional EgoSchema results in the Appendix. We believe there is a misunderstanding regarding your suggestion of MLVU; Neptune’s “known failure modes” of temporal ordering and counting are failure modes of models shown by the dataset, which already assesses those capabilities specifically. While it would always be nice to show more benchmark results, VideoMME and LongVideoBench largely consider similar capabilities to the already-presented datasets.
>
> ## 3. Validation with other base models:
> Please see note to all reviewers, where we present some additional results.
>
> ## 4. Comparison to simpler ensemble methods:
> This is a good point! A reasonable comparison point would be e.g., self-consistency from the text domain. But this actually leads to one strong motivation for CAViAR: as found by other work in video understanding (https://arxiv.org/abs/2507.06485), these methods do not appear to offer benefit in the video domain, unlike in text. We will update the paper to reflect this point.
>
> As you already noted the originality and empirical quality, we hope that given our additionally presented results and arguments, you would consider raising your score.

---

### Official Review · Reviewer_2td3 · 2025-10-30

**Soundness:** 3
**Presentation:** 3
**Contribution:** 3
**Rating:** 6
**Confidence:** 3

**Summary:**

This paper proposes a multi-agent model named CAViAR for video understanding. The model introduces two types of agents: a reasoning agent and a critic. The reasoning agent generates multiple reasoning strategies, while the critic acts as a verifier, ranking these strategies and selecting the most appropriate one. When the critic is incorporated, experimental results show that the model achieves improved performance on several popular benchmarks.

**Strengths:**

- The paper proposes CAViAR, a multi-agent–based video understanding algorithm. The method appears to be training-free, utilizing existing LLMs (Large Language Models) as agents, and designs a modular system where each agent plays a different role through prompting.


- The framework includes two types of agents: a program-generating agent and a critic. The program-generating agent interprets the video and proposes multiple reasoning strategies, while the critic ranks these strategies to determine the most appropriate one. This design enhances the model’s robustness in handling complex reasoning tasks.


- Unlike existing methods such as visual programming, which rely on a fixed reasoning procedure, the proposed algorithm allows the reasoning process to adapt dynamically based on the critic’s feedback.

**Weaknesses:**

- Lack of Novelty in the Modular System


   - As a natural limitation of this type of work, constructing a modular system purely through prompting in a training-free manner appears to lack visible novelty in terms of model architecture or training framework. Therefore, additional experiments or analyses are needed to compensate for this limitation.


- Experiments Across Diverse LLMs


   - It would be valuable to evaluate the proposed method not only with Gemini-Flash, but also with other closed-source models (e.g., ChatGPT, Claude, etc.) and open-source models, to verify whether the approach remains equally effective across different LLMs.


- Comparison of Time Consumption and Token Usage with Baselines


   - Since the proposed approach employs a multi-agent setup, it is expected to incur higher time consumption and token usage compared to baseline methods. A more detailed and quantitative analysis of these aspects is necessary.

**Questions:**

Please provide your responses with reference to the weaknesses mentioned above.

---

> ### Author Response · Authors · 2025-11-17
>
> Thank you for your positive review and recognition of our multi-agent algorithm enabling dynamic adaptation based on critic feedback. We appreciate your constructive feedback; as we believed it would be helpful for the other reviewers to also see the responses to the points you raise, we have included them all in the comment to all reviewers; please see that for detailed responses.

---

### Author Response · Authors · 2025-11-17
**Response to All Reviewers**

We thank all reviewers for their thoughtful feedback. We appreciate that reviewers recognized our contribution as a “sound framework” (Reviewer wZWB) with “clear methodological originality” (Reviewer kQk1) and “strong empirical gains” (Reviewers 2td3, kQk1, wZWB).



## 1. CAViAR Generalizes to Other Strong MLLMs

Multiple reviewers pointed out that the paper would benefit from experiments across different base LLMs. We agree with this point and have conducted preliminary quantitative experiments beyond Gemini-Flash. We implemented CAViAR with GPT-4o-mini as a base model and applied it to the recent, challenging LVBench benchmark.

| Method           | Accuracy (%) ↑ |
|------------------|----------------|
| Direct Inference (GPT-4o-mini) | 43.2           |
| **CAViAR (GPT-4o-mini)**       | **49.0**       |

(These results are obtained on a 25% subset of the total benchmark due to time, cost, and query limit considerations, but could be extended for the final submission.) We observe a substantial improvement in performance. We also explored using open-source models, but this suggested they are thus far not capable of the reasoning beyond simple video QA required for agentic approaches. We expect that as open models rapidly improve, CAViAR will work with these models as well.

## 2. Inference Compute Considerations

Reviewers also asked for additional insight into the inference overhead incurred with the agentic approach. This is an important consideration. We find each video typically requires 3-5 agent steps per strategy times 3 strategies = 9-15 total API calls. This is within the range of existing tool-based video reasoning methods: MoReVQA requires 4 reasoning calls plus perception tools per query, Video-of-Thought uses 5-6+ calls with verification re-execution, and AVIS performs 2-10+ dynamic calls with 8% requiring backtracking. Token consumption varies with video length but scales linearly with the number of frames processed in each window. While this increases compute compared to single-pass methods, the significant performance gains (15% on LVBench) justify the cost for accuracy-critical applications, consistent with the compute-accuracy tradeoffs seen across modern multi-step reasoning systems.

## 3. Comparison to VideoAgent, VideoTree, ViperGPT and Novelty

We would like to clarify the positioning of CAViAR relative to the related works the reviewers point out. While the identified works – namely, VideoAgent, VideoTree, and ViperGPT – are indeed closely related, CAViAR has key differences to each that lead to significant performance gains over prior art, as we show in our ablations.

VideoAgent follows a fixed iterative procedure (caption, retrieve, check confidence, repeat) rather than dynamically choosing tools based on reasoning needs. CAViAR's agent adaptively selects tools based on intermediate results, making it fundamentally different from VideoAgent's rigid pipeline. The primary similarity lies in how both methods terminate: CAViAR ends trajectories based on the agent choosing a tool indicating completion, then selects the most promising with the critic, while VideoAgent asks the base LLM used how confident it is in the final answer, terminating when confident.

We compare with VideoAgent through the ablation in Table 5, which implements VideoAgent's key confidence-based iteration mechanism. Our results show the VideoAgent self-confidence based approach actually hurts performance (39.9% vs 47.1% for the agent alone) while our critic enhances it (62.0%).

We also compare to the single-program approach – equivalent to ViperGPT – in Table 4, finding this baseline  yields a 20% drop in performance relative to the agent alone and a 35% drop compared to the agent and critic together, highlighting the importance of the core contributions of CAViAR.

VideoTree focuses on frame selection rather than general video reasoning with diverse tools. A direct comparison would require reimplementing our full tool suite within their frameworks, i.e., using their frame selection approach as part of our agentic pipeline, which is beyond scope but an interesting direction for future work.

CAViAR is thus the first work to demonstrate success for agentic reasoning in video as it is commonly defined for current LLM agents: using tools sequentially to achieve a goal until choosing to stop. While there has been prior work also working towards modular, training-free systems, we consider this a paradigm in its own right given the breadth of potential approaches it entails.

---

### Meta-Review · Area_Chair_vd8B · 2026-01-06

**Summary:**

This paper investigates agentic video reasoning by enabling a large language model to dynamically orchestrate video modules as tools, guided by a critic that distinguishes successful from unsuccessful reasoning trajectories. The motivation is timely, given the growing difficulty of long-video and complex reasoning benchmarks.

However, key concerns remain unresolved. In particular, the paper does not verify the proposed approach on more recent and widely adopted benchmarks such as Video-MME, as requested by Reviewer kQk1-W2 & Q2 and Reviewer wZWB-W3, which limits the strength of the empirical evidence and the relevance of the reported gains to current evaluation standards. In addition, multiple reviewers (Reviewer 2td3-W1, Reviewer kQk1-W1, Reviewer wZWB-W1) raised concerns regarding the relatively weak novelty of the proposed agentic framework, noting its close resemblance to prior agent-based and tool-using approaches in both vision and language reasoning.

As a result, despite promising motivation and results on selected benchmarks, the paper does not sufficiently address the reviewers’ central concerns regarding evaluation breadth and methodological novelty.

**Reviewer Concerns:**

Concerns regarding novelty (Reviewer 2td3-W1, Reviewer kQk1-W1, Reviewer wZWB-W1) are shared across reviewers and remain insufficiently resolved. In addition, concerns about evaluation on more recent and comprehensive benchmarks (Reviewer kQk1-W2, Q2; Reviewer wZWB-W3) also remain unaddressed.

**Reviewer Scores:**

I expect all three reviewers to maintain their current ratings due to the remaining unresolved concerns.

---

### Decision · Program_Chairs · 2026-01-26

Reject